**Data Availability Statement:** All relevant data are within the manuscript and its Supporting Information files (ZIP FILE).

# Evaluation of anticancer potential of tetracene-5,12-dione (A01) and pyrimidine-2,4-dione (A02) via caspase 3 and lactate dehydrogenase cytotoxicity investigations

Mubashir Aziz[1], Muhammad Sarfraz[2], Muhammad Khurrum Ibrahim[3], Syeda Abida Ejaz[1]*, Tasneem Zehra[4], Hanan A. Ogaly[5], Mosab Arafat[2], Fatimah A. M. Al-Zahrani[5], Chen Li[6]*

1 Department of Pharmaceutical Chemistry, Faculty of Pharmacy, The Islamia University of Bahawalpur, Bahawalpur, Pakistan, 2 College of Pharmacy, Al Ain University, Al Ain, United Arab Emirates, 3 Baqai Medical University, Karachi, Pakistan, 4 Department of Basic Science & Humanities, Dawood University of Engineering & Technology, Karachi, Pakistan, 5 Chemistry Department, College of Science, King Khalid University, Abha, Saudi Arabia, 6 Department of Biology, Chemistry, Pharmacy, Free University of Berlin, Berlin, Germany

* abida.ejaz@iub.edu.pk (SAE); chen.li@fu-berlin.de (CL)

## Abstract

Cancer stands as a significant global cause of mortality, predominantly arising from the dys-regulation of key enzymes and DNA. One strategic avenue in developing new anticancer agents involves targeting specific proteins within the cancer pathway. Amidst ongoing efforts to enhance the efficacy of anticancer drugs, a range of crucial medications currently interact with DNA at the molecular level, exerting profound biological effects. Our study is driven by the objective to comprehensively explore the potential of two compounds: (7S,9S)-7-[(2R,4S,5S,6S)-4-amino-5-hydroxy-6-methyloxan-2-yl]oxy-6,9,11-trihydroxy-9-(2-hydroxyacetyl)-4-methoxy-8,10-dihydro-7H-tetracene-5,12-dione (A01) and 5-fluoro-1H-pyrimidine-2,4-dione (A02). These compounds have demonstrated marked efficacy against breast and cervical cancer cell lines, positioning them as promising anticancer candidates. In our investigation, A01 has emerged as a particularly potent candidate, with its potential bolstered by corroborative evidence from lactate dehydrogenase release and caspase-3 activity assays. On the other hand, A02 has exhibited remarkable anticancer potential. To further elucidate their molecular mechanisms and interactions, we employed computational techniques, including molecular docking and molecular dynamics simulations. Notably, our computational analyses suggest that the A01-DNA complex predominantly interacts via the minor groove, imparting significant insights into its mechanism of action. While earlier studies have also highlighted the anticancer activity of A01, our research contributes by providing a deeper understanding of its binding mechanisms through computational investigations. This knowledge holds potential for designing more effective drugs that target cancer-associated proteins. These findings lay a robust groundwork for future inquiries and propose that derivatives of A01 could be synthesized as potent bioactive agents for cancer

**Funding:** The authors extend their appreciation to the Deanship of Scientific Research (DSR), King Khalid University, Abha, Saudi Arabia, under Grant No. RGP.2/93/44. The funders had no role in study design, data collection and analysis, decision to publish, or preparation of the manuscript.

**Competing interests:** The authors have declared that no competing interests exist.

treatment. By elucidating the distinctive aspects of our study's outcomes, we address the concern of distinguishing our findings from those of prior research.

## 1. Introduction

Cancer is a multifaceted disease that is characterized by the uncontrollable growth and subsequent dissemination of abnormal cells throughout the body. Given that it is consistently ranked among the leading causes of death across the globe, it presents a significant challenge to the field of medicine [1]. Traditional treatments for cancer, such as chemotherapy and radiation therapy, are not only ineffective but also come with a host of unpleasant side effects [2]. As a direct consequence of this, there is a growing interest in the development of innovative and efficient cancer treatments. Targeting specific proteins that are involved in the cancer pathway is one strategy that can be used in the process of creating new anticancer medications [3]. Cancer is the second leading cause of death around the world, and while dysfunctional DNA can contribute to its development, it is not the sole cause. Other factors such as genetics, environmental exposures, altered cellular pathways, proteins and transcription factors, have been associated with the progression, differentiation and development of different kinds of cancer [4]. It is believed that certain medicines can kill cancer cells or cause them to shrink [5]. It is widely believed that inhibiting the ability of cancer cells to replicate their DNA would be the most effective treatment [6]. In contrast, medicines that bind DNA reversibly pose a much lower risk to healthy cells and are therefore preferable for therapeutic use [7]. This emphasizes the sustained importance of research into chemical compounds that can form reversible bonds with DNA and prevent cancer without harming normal cells. The incorporation of non-covalent interactions such as electrostatic interactions, groove binding, and intercalation into in vitro studies evaluating the anticancer potential of drug-like molecules in terms of binding modes and properties [8] is of considerable benefit. Caspases 3, p53, and NF-κB are potential therapeutic targets for various types of cancers, including breast cancer, lung cancer, leukemia, lymphoma, colon cancer, ovarian cancer, prostate cancer, and hematologic malignancies like leukemia, lymphoma, and multiple myeloma [9–14]. Caspase 3 is involved in the programmed cell death or apoptosis. Dysfunctional caspase 3 has been associated with the development of cancer, including breast cancer, lung cancer, leukemia, and lymphoma [10]. P53 is a tumor suppressor protein that regulates the cell cycle and prevents the growth of cancer cells. Mutated or dysfunctional p53 has been linked to various types of cancer, including breast cancer, lung cancer, colon cancer, and ovarian cancer [11, 12, 15]. The nuclear factor-kappa B (NF-κB) pathway is involved in the regulation of immune response and inflammation, cell survival, and apoptosis [13]. Dysregulated NF-κB signaling has been associated with the development and progression of various types of cancer, including breast cancer, prostate cancer, and hematologic malignancies such as leukemia, lymphoma, and multiple myeloma [14]. Therefore, these proteins are being considered as emergent targets for the designing of anticancer agents.

Heterocyclic compounds are one of the many chemical classes that play an important role in medicinal chemistry and the rational design of new drugs [16, 17]. It has been demonstrated that numerous heterocyclic compounds are capable of exhibiting a diverse array of anticancer activities [18, 19], including the inhibition of DNA replication, the induction of apoptosis, and the disruption of cell signaling pathways. Compounds based on pyrimidine and pyrazole, for instance, have been demonstrated to exhibit potent anticancer activities through a variety of mechanisms [20]. These mechanisms include the inhibition of cell proliferation, the induction

of apoptosis, and the disruption of cell signaling pathways. For instance, it has been demonstrated that 1,3,5-trisubstituted pyrazole derivatives have antiproliferative activity against a variety of cancer cell lines [21]. In addition, it has been demonstrated that compounds based on pyrazole inhibit the involved in the process of repairing DNA and possessed anti proliferative activities [22]. Based on findings, the current study has employed two heterocyclic compounds for investigation as potential anticancer candidate. Compound A01 is a derivative of doxorubicin which is a well-known and widely used chemotherapy drug for the treatment of various types of cancer. Doxorubicin works by interfering with the DNA in cancer cells, preventing them from reproducing and causing them to die. The derivative mentioned in the question is a modified form of doxorubicin, which has been designed to enhance its effectiveness and reduce its toxicity. Studies have shown that the modified compound has improved antitumor activity compared to doxorubicin, while also being less toxic to healthy cells. This makes it a promising candidate for further investigation as a potential chemotherapy drug for the treatment of cancer. Furthermore, the compound has been shown to be effective against a wide range of cancer types, including breast cancer, lung cancer, and leukemia [23]. Furthermore, compound A02 is a pyrimidine analog that has been shown to have anti-tumor activity in various types of cancer. It is a fluorinated analog of uracil and inhibits the synthesis of thymidylate, which is necessary for DNA replication in cancer cells. This leads to the inhibition of cancer cell proliferation and ultimately results in cell death [24]. These findings provide rationale for selecting these heterocyclic compounds (A01 and A02) for the development of advanced medicines that are highly effective in treatment against cancer disease.

In addition, different findings has suggested that caspase 3, p53, and NF-κB are proteins that show potential as therapeutic targets in the development of new anticancer drugs against breast cancer, lung cancer, leukemia, lymphoma, colon cancer, ovarian cancer, prostate cancer, and hematologic malignancies like leukemia, lymphoma, and multiple myeloma [9, 15–18, 25, 26]. The continuation of research and development efforts in these areas holds a great deal of promise for enhancing cancer treatment and the outcomes for cancer patients. Few previously reported pyrazole and pyrimidine analogues exhibiting anticancer potential are illustrated in the Fig 1 [27–30].

The current study has examined the potential anticancer activity of two derivatives, (compound A01) and (compound A02), using a combination of in vitro assays, computational methods, and molecular simulations. Both compounds were evaluated for their impact on cancer cell lines and DNA binding capacity utilizing Hearing sperm DNA. DFT calculations were performed to assess electronic structure and compound stability, while molecular docking studies were employed to investigate the binding affinity of the compounds with caspase-3, p53, and kappa protein. Furthermore, MD simulations was conducted to analyze the stability of the protein-ligand complexes. The results of this study have the potential to inform the development of future anticancer therapies, with both (compound A01) and (compound A02) showing promise as potent therapeutic agents.

## 2. Experimental

### 2.1. Cell viability assay

In this study, the assessment of the anticancer potential of two compounds, A01 and A02, was conducted against human breast cancer cell lines from ATTC (MDA-MB231: HTB-26™ and MCF-7 cell line: HTB-22™) and a human cervical cancer cell line (HeLa: CRM-CCL-2™). The cell viability assay was performed as described previously [32]. The detailed description is given in S1 File.

**4-bromo-1-(phenylsulfonyl) pyrazole**

**(7S,9S)-7-[(2R,4S,5S,6S)-4-amino-5-hydroxy-6-methyloxan-2-yl]oxy-6,9,11-trihydroxy-9-(2-hydroxyacetyl)-4-methoxy-8,10-dihydro-7H-tetracene-5,12-dione (A01)**

**3-[(2S,3R,4S,5R)-3,4-dihydroxy-5-(hydroxymethyl)oxolan-2-yl]-4-hydroxy-1H-pyrazole-5-carboxamide**

**5-fluoro-1H-pyrimidine-2,4-dione (A02)**

**3-[(1R)-1-(2,6-dichloro-3-fluorophenyl) ethoxy]-5-(1-piperidin-4-ylpyrazol-4-yl) pyridin-2-amine**

**Fig 1. Already reported Pyrazole derivatives alongside compound A01 and A02 [28, 30, 31].**

## 2.2. Lactate dehydrogenase cytotoxicity assay

Lactate dehydrogenase for the cytotoxicity potential of **A01** and **A02** against HeLa, MDA-MB-231 and MCF-7 cells was determine at its respective $GI_{50}$ and 2x $GI_{50}$ values. The selection of HeLa, MDA-MB-231, and MCF-7 cell lines for anticancer studies is based on their aggressive and invasive nature, resistance to some chemotherapeutic agents, and their widespread use in cancer research. In addition, The primary objective of this study was to explore promising leads for combatting specific types of cancer, particularly breast and cervical cancers. Extensive research has established MCF-7 and MDA-MB-231 as breast cancer cell lines, while the HeLa cell line is associated with cervical metastasis. These cell lines are valuable tools for investigating cancer metastasis and for developing new cancer treatments. The previously reported method was used [33] and as per protocol mentioned in LDH Assay Kit (Cytotoxicity; ab65393). The absorbance was carried out at CLARIOstar Plus microplate reader (BMG Lab-tech, Germany). The detailed description is given in S1 File.

## 2.3. Apoptosis assessment by caspase 3 activity

The method used for treatment was in accordance with literature [34] and the Caspase-3 Assay Kit protocol (Fluorometric: ab39383). The adherent cells were collected, centrifuged, and lysed on ice for 10 minutes using 50 μL of lysis buffer. The lysate was then incubated with DEVD-AFC substrate for caspase-3, followed by the addition of reaction buffer at 37˚C for 3 hours. The amount of fluorescent cleavage product was measured using the CLARIOstar Plus microplate reader (BMG Labtech, Germany). The experiments were conducted in triplicate. More detailed information can be found in the S1 File.

## 2.4. Spectrophotometric DNA binding analysis

UV-visible spectrophotometry was utilized at room temperature to examine the interaction of (compound A01) and (compound A02) with DNA when bound. Initially, both compounds were prepared in 10% DMSO. A stock solution was made by dissolving 5 mg of lyophilized Herring sperm DNA (Sigma Aldrich, USA) in 10 mL of distilled water. To test the DNA's purity, we calculated the ratio of absorbance at 260 and 280 nm, obtaining values ranging from 1.6 to 1.9. Various concentrations of HS-DNA (40 μM, 80 μM, 120 μM, 160 μM, 200 μM, and 240 μM) were employed in the experiment. More detailed information is available in the methodology section [35, 36] in both the absence and presence of the compounds. The UV absorption spectra was recorded using a CLARIOstar Plus microplate reader (BMG Labtech, Germany) after a 30-minute incubation in the dark at room temperature.

## 2.5. Computational investigations

**2.5.1 Density functional theory calculations.** The current study have utilized the Gaussian 09W program [37, 38] to assess the structural stability and precision of the compound **A01** and **A02** through the optimization of their structural geometries and frequency calculations. The electron density, frontier molecular orbitals, and both local and global reactivity descriptors for the compounds were computed to acquire a more in-depth understanding of their electron density [39]. Quantifying the electron density of the compounds allowed for determination of their reactivity profiles, revealing a high ionization potential and chemical adaptability. Density functional theory (DFT) was implemented to calculate the electron density using the 6-31G* basis set and the B3LYP functional correlation, as detailed in the methods section [40]. This basis set is a hybrid of the 6-31G and 3-21G sets, incorporating diffuse functions necessary for modeling electron density near the periphery of molecules while maintaining a balance between accuracy and computing efficiency [41]. To accurately and effectively characterize the electrical properties of molecules, the selected basis set consists of primitive and Gaussian orbitals. GaussView 06 [42] log files were analyzed to generate a number of analytical metrics, which were then used to evaluate the electronic properties of the compounds.

**2.5.2. Molecular docking studies.** To analyze the non-covalent interactions between (A01) and (A02) and numerous anti-cancer proteins and nucleic acids, proteins of interest were retrieved from the Protein Data Bank (PDB IDs 3DEI, 1NFI, 3DCY, and 127D; www.rcsb.com). Macromolecules were then dehydrated using MGL techniques, with heteroatoms being replaced by polar hydrogen atoms and Gasteiger charges introduced to prepare for molecular docking [43]. The accurate binding pattern of a ligand is reliant on the protonation state of the enzyme's active site. In order to ensure a precise reflection of the proteins, MGL correction tools were employed to remedy any incorrectly absent residues. Proper preparation was then conducted to facilitate interaction of each protein. The docking database was constructed, with optimization of the ligand via density functional theory calculations being done to enable docking preparations for both molecules. The compounds were saved using the

pdbqt format. AutoDock Vina was utilized to perform molecular docking after the protein and ligand databases were properly prepared [44, 45]. Following molecular docking, the proposed binding sites of all proteins were selected based on functional significance and previously reported literature [46]. The coordinates for caspase-3 were (-46.790, 15.0200, -21.901), for NF-κB they were (-10.249, 48.903, and 6.706), and for p53, the coordinates were (30.483, 32.903, and -2.903 Regarding the DNA molecule, we included the entire molecular structure within the designated grid box for the purpose of molecular docking. To ensure that only accurate docking predictions were provided, the number of modes was restricted to 100. Following docking evaluations, the optimal conformations were investigated using QM/MM.

The validity check is essential for establishing the reliability of the molecular docking technique. To validate the docking method, the co-crystallized ligand (a reference inhibitor) was redocked at native site. A docking was determined to be valid if the root-mean-square deviation (RMSD) between the native and regenerated posture was less than 2 angstroms. The validity check is essential for establishing the reliability of the mooring technique. To validate the docking method, the co-crystallized ligand (a reference inhibitor) was redocked. The RMSD between the original and regenerated posture must be less than 2 angstroms in order to validate the mooring technique [47]. The redocked pose of co-crystal ligand is provided in **S1 File** (**S1 Fig**).

**2.5.3. Molecular dynamics simulations.** The protein-ligand complex generated by molecular docking was subjected to molecular dynamics simulations based on Desmond [48]. The TIP3P solvent model was utilized to simulate each complex for 50 ns [49]. To accomplish neutralization of the system, 0.15M sodium chloride (NaCl) ions were added. The simulation utilized the OPLS3 (optimal potential for liquid simulation 3) [50] forcefield, which accounted for the mobility of the atoms within regularly spaced boundaries. In order to prevent atomic collisions, a preliminary energy reduction technique consisting of 2000 steps was put into action. At a temperature of 300 kelvin and a pressure of 1.01 bar, the system was brought to equilibrium within of an isothermal and isobaric (NPT) ensemble [51]. A cutoff distance of 10 angstroms was used for the study of short-range van der Waals interactions. For the purposes of simulation, we made use of the Nose-Hoover thermostat in conjunction with the Martyna-Tobias-Klein barostat [52]. Integrating the equations of motion with a time step of 2 fs at each step. The manufacturing cycle lasted for fifty nanoseconds, and the simulated trajectories were stored in intervals that were fifty picoseconds long. The electrostatic interactions were analyzed in great depth by making use of the Particle Mesh Ewald approach [53]. Analyses of the simulated trajectories of protein-ligand complexes were carried out by employing the Desmond simulation interaction diagram approach.

**2.5.4 *ADMET* prediction.** ADME prediction plays a pivotal role in the realm of drug design research by facilitating the enhancement of both the effectiveness and safety profiles of novel compounds. Among the array of tools available for predicting ADME properties, Swiss ADME stands out as a widely embraced choice [54]. In the present study, the assessment of ADME characteristics for both compounds, A01 and A02, was undertaken through the utilization of the Swiss ADME online platform. The workflow commenced with the formulation of input files, encompassing the generation of SMILES representations for each compound. Subsequently, the Swiss ADME tool was employed to analyze the input files, thereby yielding a comprehensive dataset. The ensuing phase encompassed a meticulous analysis of the generated outcomes, enabling the derivation of meaningful insights into the ADME properties inherent to the compounds under investigation. This systematic process culminated in the formulation of well-founded conclusions concerning the ADME behavior of the aforementioned compounds.

## 3. Result and discussions

### 3.1. Biological evaluation

**3.1.1. *In vitro* cytotoxic activity.**   In the current study, an in vitro assay was utilized for evaluation of both compounds A01 and A02 against three cancer cell lines (HeLa, MCF-7, and MDA-MB-231), with a colorimetric MTT assay being performed [32, 55]. The reference compound for this cytotoxic evaluation was cisplatin. Table 1 contains the results. The $GI_{50}$ graphs have been provided in the **S1 File** **(S2 Fig).**

**3.1.2. Lactate dehydrogenase cytotoxicity assay.**   LDH is a cytosolic enzyme that catalyzes the transformation of l-lactate to pyruvate. Leakage of LDH from the cytoplasm into the medium indicates a change in the permeability of the plasma membrane or the occurrence of apoptosis or necrosis. To determine the effect of derivative A01 on LDH activity, HeLa, MDA-MB231, and MCF-7 cells were treated for 24 hours with varying concentrations of derivative A01 (Fig 2). After 24 hours of treatment with higher concentrations of compound A01, the results demonstrated a statistically significant induction of LDH (P<0.05).

**3.1.3. Apoptosis assessment by caspase 3 activity.**   Caspase-3 is the primary executor of apoptotic cell death, and its presence enhances the effectiveness of cell death. In the LDH experiment, HeLa, MDA-MB231, and MCF-7 cells were treated at two distinct concentrations with compound A01. Fig 3 demonstrates that both concentrations significantly increased caspase-3 activity. HeLa, MDA-MB231, and MCF-7 cell inhibition was dose-dependent. Both the MTT test and the LDH assay demonstrated a connection between HeLa and MDAMB-231, but only the MTT test demonstrated a significant effect.

**3.1.4. UV-visible spectroscopy based DNA binding studies.**   A01 and A02 were tested for their interactions with mammalian DNA (HS-DNA) in order to better understand their molecular behavior. The absorption spectra of both compounds were determined at a constant concentration of 10μ M (each) in the absence and presence of HS-DNA at 40 μM, 80 μM, 120 μM, 160 μM, 200 μM, and 240 μM. Figs 4 and 5 illustrate that increases in HS-DNA concentration were accompanied by increases in absorbance for both A01 and A02, but no alterations in band positions. The hyperchromic result therefore disclosed the non-covalent character of the HS-DNA interaction with both compounds. Surprisingly, A01 had the maximum DNA interaction Gibbs free energy of -18.28 KJ/mol. Nonetheless, the second chemical, i.e., also demonstrated positive results, albeit with fewer binding interactions, lending credibility to the cell viability findings.

### 3.2. Density functional theory calculations

Utilizing optimization and frequency calculations at the B3LYP/6-31G* level of theory, the structural geometries of compounds A01 and A02 were determined. The structural geometries of compounds A01 and A02 were tailored to attain the sharpest energy gradient and zero imaginary frequency. The compounds' electronic properties were also analyzed. Table 2 displays the optimization and reactivity properties of the compounds.

In this study, we analyzed the properties of two compounds, A01 and A02, which were optimized using a computational approach. The optimization energy, which reflects the stability of the molecule, was found to be -1927.951181 hartree for **A01** and -513.885145 hartree for **A02**. We also examined the polarizability of the compounds, which is a measure of how easily the electron cloud can be distorted. The polarizability values were 352.777289 and 55.367297 atomic units for A01 and A02, respectively. The dipole moment, which indicates the distribution of electric charge within the molecule, was 5.314323 debye for **A01** and 4.296947 debye for **A02**. We further investigated the electronic properties of the compounds, including the

**Table 1. Cytotoxic activity of compounds A01 & A02 against human HeLa, MDA-MB- 231 and MCF-7 cancer cell lines.**

| Compound | $GI_{50}\pm SEM$ | | | %Growth inhibition |
|---|---|---|---|---|
| | HeLa | MDA-MB231 | MCF-7 | Vero cells |
| A01 | 3.64± 0.19 | 10.4± 0.94 | 16.3± 0.96 | 6.31% |
| A02 | 27.8± 1.76 | 54.3± 3.11 | 26.8± 1.36 | 8.2% |
| Doxorubicin | 4.21± 0.22 | 6.82± 0.59 | 7.32± 0.81 | 6.67% |
| Cisplatin | 2.64± 0.13 | 2.27± 0.24 | 4.63± 0.21 | 4.88% |

Table 1 shows that the compound A01 (a) was the most active against HeLa with $GI_{50}$ values of 3.64±0.19 μM. This compound showed 10.4±0.94 and 16.3± 0.96 μM against MDA-MB231 and MCF-7, respectively. Interestingly, the compound a showed comparable results with reference drugs. Whereas compound A02 (b), compared to N-A01 (a), exhibited less potential of growth reduction of all cells i.e., HeLa, MDA-MB231 and MCF-7 cell with an $GI_{50}$ values of 27.8± 1.76, 54.3± 3.11 and 26.8± 1.36 μM, respectively. The results were further supported by DNA studies and suggested these compounds as lead for the synthesis of more potential anticancer agents.

ionization potential and electron affinity. The ionization potential reflects the energy required to remove an electron from the molecule, while the electron affinity measures the energy change when an electron is added to the molecule. Compound **A01** had an ionization potential of 0.1155 eV and an electron affinity of 0.220 eV, while compound **A02** had an ionization potential of 0.06599 eV and an electron affinity of 0.068 eV. In addition, we examined the electron-donating and accepting powers of the compounds, as well as their electrophilicity. These properties are important for understanding the reactivity of a molecule and its potential for interacting with other compounds. Compound A01 had a higher electron-donating power (ω-) of 0.378 and a higher electron-accepting power (ω+) of 0.599 compared to A02, which had ω- and ω+ values of 0.231 and 0.298, respectively. Compound A01 also had a higher electrophilicity (Δω±) of 0.271 compared to A02, which had a Δω± value of 0.100. The properties analyzed in this study are important for understanding the behavior and potential applications of these compounds in various fields, such as drug discovery and materials science. The optimization energy, polarizability, and dipole moment are essential for evaluating the stability and reactivity of a molecule, while the ionization potential and electron affinity can provide insights into its electronic properties. The electron-donating and accepting powers and electrophilicity are particularly relevant for understanding the interactions between different molecules and their potential for forming bonds.

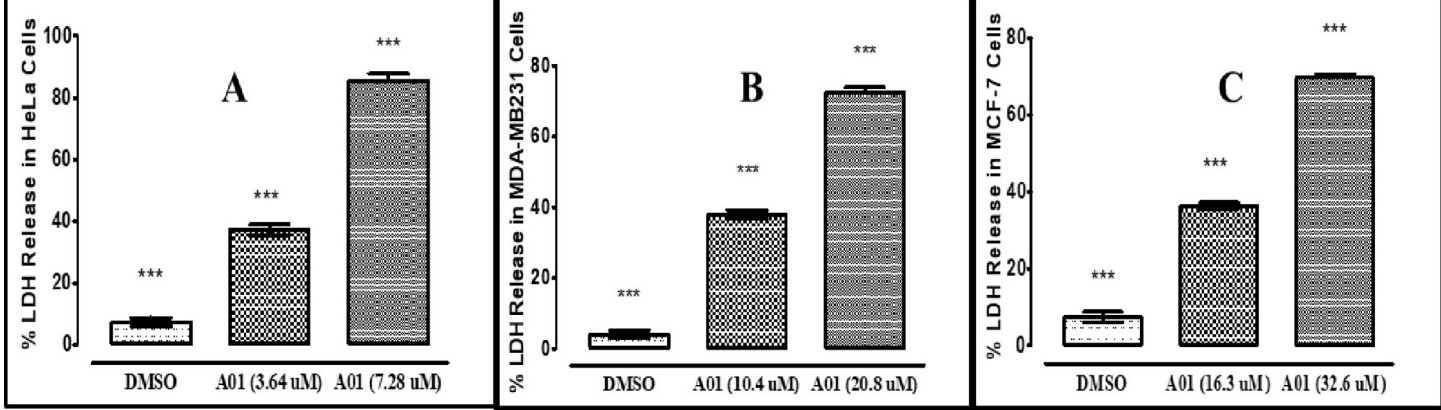

**Fig 2.** The representation of % Lactate dehydrogenase (LDH) release at different concentration of A01 against HeLa (A), MDA-MB-231 (B) and MCF-7 cells using LDH Assay Kit (Cytotoxicity) (ab65393). Data represent the mean ± SD (n = 3). * p < 0.05 compares the treated cell with control cell.

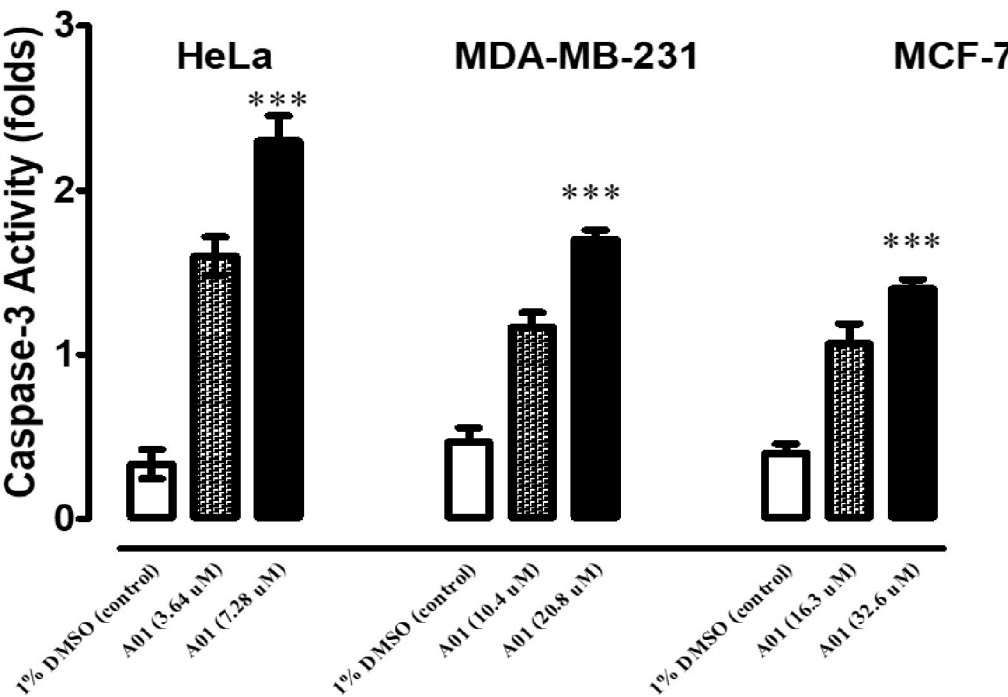

**Fig 3. Detection of Caspase-3 activity was detected in HeLa, MDA-MB231 and MCF-7 cells treated at two different concentrations, respectively, for 24 h using Caspase-3 Assay Kit (Fluorometric:ab39383).** Data represent the mean ± SD (n = 3). * p < 0.05 compares the treated cell with control cell.

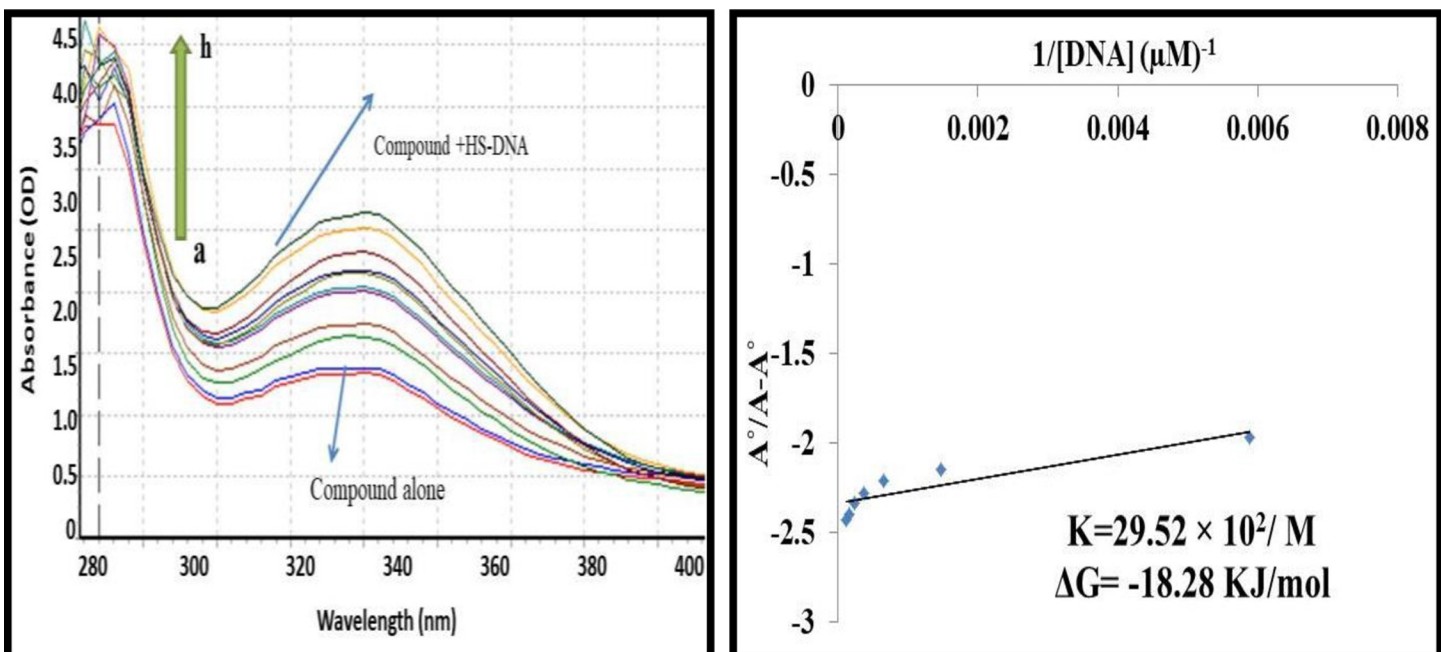

**Fig 4.** A) UV-visible spectrum responses of **A01** in the absence and presence of various concentrations of HS-DNA as was highlighted above. The pointed end of the arrow represents an increase in the amount of DNA present. (B) Ao–A/Ao vs. 1/[DNA] is the graph that is used to calculate the binding constant.

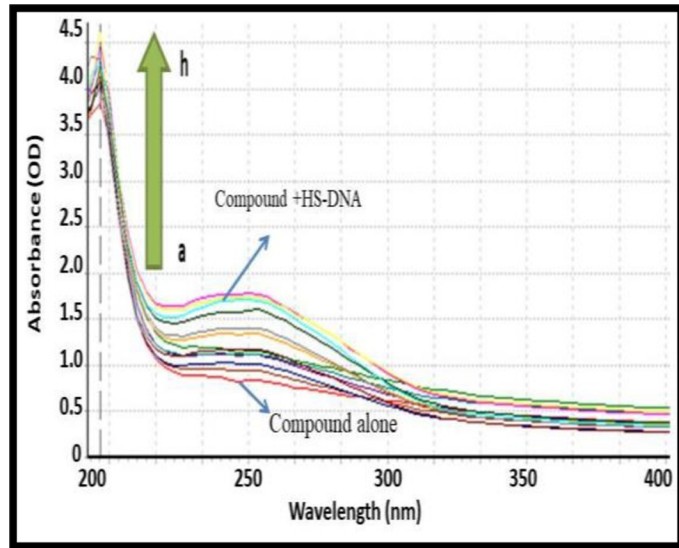
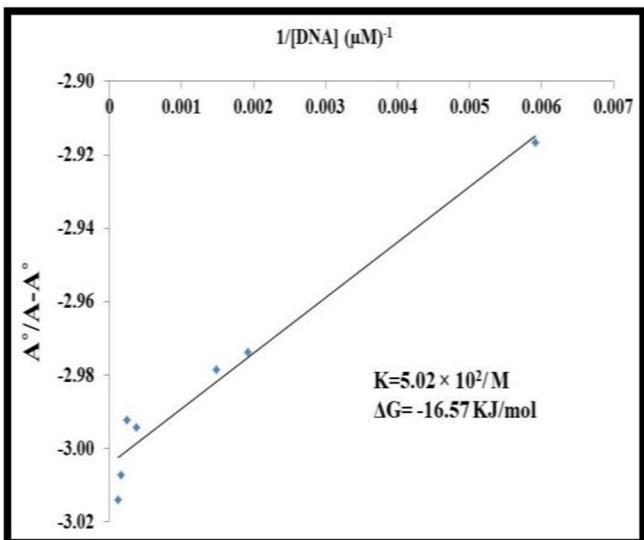

**Fig 5.** A) UV-visible spectrum responses of **A02** in the absence and presence of various concentrations of HS-DNA as was highlighted above. The pointed end of the arrow represents an increase in the amount of DNA present. (B) Ao–A/Ao vs. 1/[DNA] is the graph that is used to calculate the binding constant.

The electrostatic potential map depicts the molecule's high- and low-potential regions. Due to the presence of electronegative oxygen atoms, red indicates an electron-attracting potential, whereas blue indicates a low electron-donating potential due to the presence of hydrogen atoms. These diagrams teach us about the properties and actions of a molecule by illustrating where the electrons are and how they react. Fig 6 illustrates the ESP map and optimized structures for compounds A01 and A02.

In this study, two compounds, A01 and A02, were analyzed to determine their electronic properties. For compound A01, the EHOMO and ELUMO are -0.2003 eV and -0.115 eV, respectively, resulting in a $\Delta$Egap of 0.084 eV. The chemical hardness ($\eta$) is calculated as 0.042 eV, indicating that the compound is relatively soft and more reactive. The chemical potential ($\mu$) is -0.158 eV, while the electrophilicity index ($\omega$) is 0.294 eV, which suggests that the compound is a good electron donor and a weak electrophile. The chemical softness (S) is 11.792 eV, indicating that the compound has a low resistance to deformation. The electronegativity (X) is 0.158, which reflects the ability of the compound to attract electrons towards itself.

For compound A02, the EHOMO and ELUMO are -0.2608 eV and -0.065 eV, respectively, resulting in a $\Delta$Egap of 0.194 eV. The chemical hardness ($\eta$) is calculated as 0.097 eV, indicating that the compound is relatively harder and less reactive than A01. The chemical potential ($\mu$) is -0.163 eV, while the electrophilicity index ($\omega$) is 0.137 eV, which suggests that the compound is a weak electron donor and a good electrophile. The chemical softness (S) is 5.133 eV, indicating that the compound has a higher resistance to deformation than **A01**. The electronegativity (X) is 0.163, which reflects the ability of the compound to attract electrons towards itself, similar to **A01**. These properties can provide valuable insight into the potential

**Table 2. Optimization energy, polarizability and dipole moment of compound A01 and A02.**

| Code | Optimization Energy (hartree) | Potential Ionization I(eV) | Affinity A (eV) | Electron donating power ($\omega$-) | Electron accepting Power ($\omega$+) | Dipole Moment (debye) | Electro philicity ($\Delta\omega\pm$) | Polarizability a.u ($\alpha$) |
|------|------|------|------|------|------|------|------|------|
| **A01** | -1927.951181 | 0.2003 | 0.1155 | 0.220 | 0.378 | 5.314323 | 0.599 | 352.777289 |
| **A02** | -513.885145 | 0.2608 | 0.06599 | 0.068 | 0.231 | 4.296947 | 0.298 | 55.367297 |

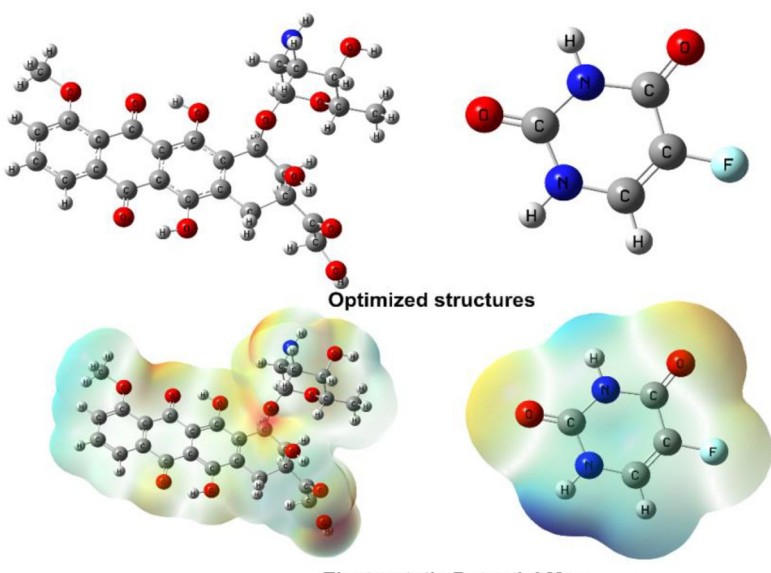

**Fig 6.** The optimized structure A01 (**right**) and A02 (**left**) alongside electrostatic potential map (ESP).

applications of these compounds in various fields. The global and local reactivity descriptor values for compound A01 and A02, is tabulated in Table 3.

The hydrophobic benzene ring and methyl group of compound A01 contribute a small quantity to the HOMO, whereas the sulphonamide group is responsible for the majority of the HOMO. Due to the clustering of LUMOs surrounding the toluene ring, this molecule is electrophilic. The dichloro-substituted thiophene ring of compound A02 contains an abundance of HOMO orbitals. Concentrated LUMO orbitals are present in the sulphonamide portion of the molecule. These results shed light on the molecules' chemical reactivity and electrical structure. The FMOs of compounds A01 and A02 are depicted in Fig 7.

## 3.3. Molecular docking studies

The objective of the current investigation was to assess the DNA intercalation property of two compounds, A01 and A02 against three important cancer-related proteins, caspase-3, NF-κB, and p53. These proteins are essential for the progression of cancer, and inhibiting their activity is a crucial stage in preventing tumor growth. Stress-induced apoptosis is mediated by Caspase-3, whereas NF-κB regulates inflammation, cell survival, and proliferation. In contrast, P53 functions as a tumor suppressor by modulating the cell cycle and is known as the "guardian of the genome."

Caspase 3 is a cysteine protease that plays a crucial role in apoptosis, or programmed cell death. Its topology can be described as a helix bundle, with six alpha-helices arranged in a compact fold. The protein is composed of two subunits, each containing a large and a small domain.

**Table 3. Global and local reactivity descriptors of compound A01 and A02.**

| Compound | $E_{HOMO}$ (eV) | $E_{LUMO}$ (eV) | $\Delta E_{gap}$ (eV) | Chemical Hardness ($\eta$) | Chemical Potential ($\mu$) | Electrophilicity Index ($\omega$) | Chemical Softness (S) | Electronegativity (X) |
|---|---|---|---|---|---|---|---|---|
| A01 | -0.2003 | -0.115 | 0.084 | 0.042 | -0.158 | 0.294 | 11.792 | 0.158 |
| A02 | -0.2608 | -0.065 | 0.194 | 0.097 | -0.163 | 0.137 | 5.133 | 0.163 |

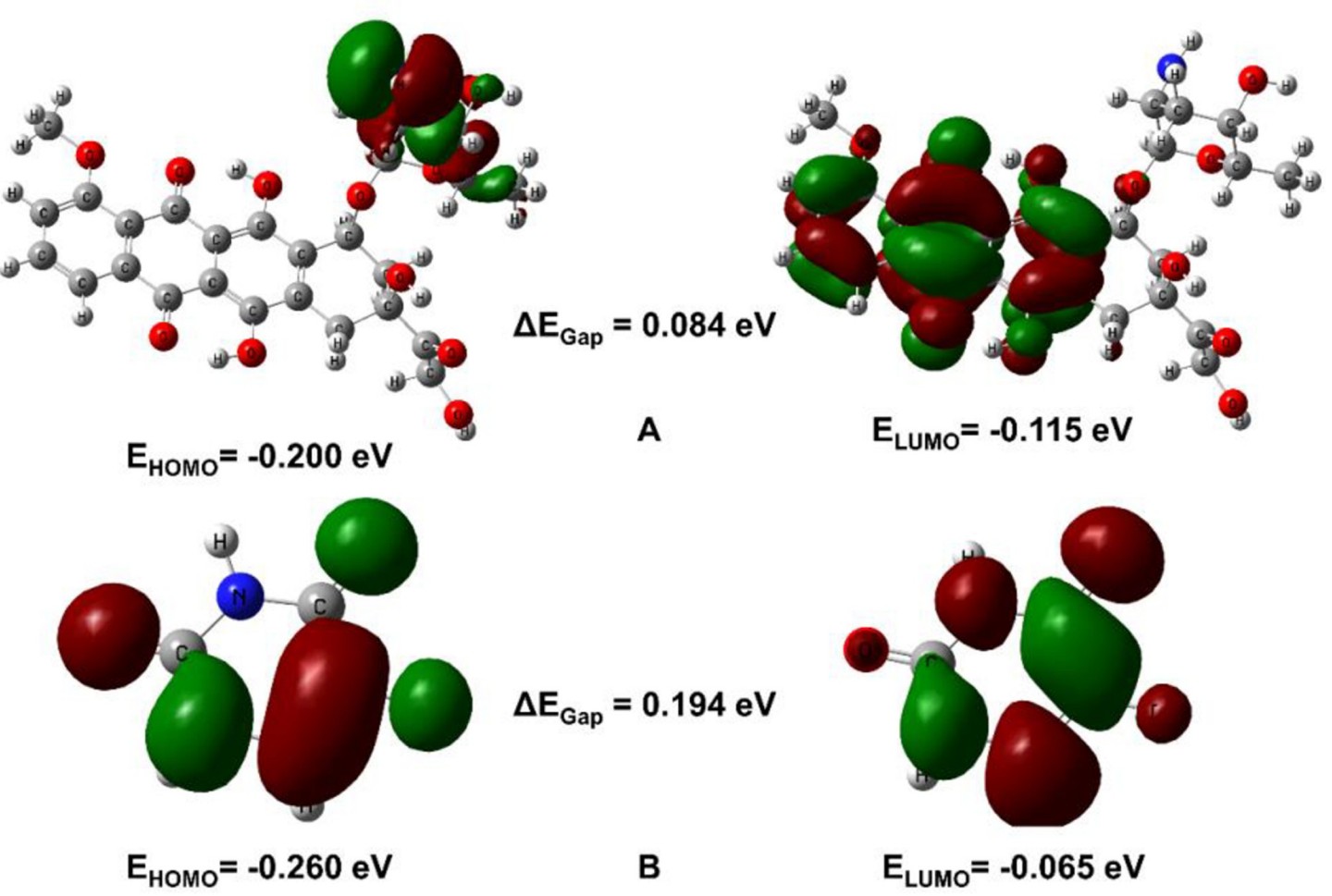

**Fig 7.** HOMO/LUMO orbitals of A01 (**up**) and A02 (**down**).

The large domain is responsible for substrate binding, while the small domain is involved in catalysis. The ternary structure of caspase 3 is composed of three distinct domains: the N-terminal prodomain, the large and small catalytic subunits. The prodomain is important for regulating the activation of caspase 3, while the catalytic subunits are responsible for catalyzing the hydrolysis of specific peptide bonds. The active site contains a cysteine residue that plays a crucial role in the catalytic mechanism of caspase 3, and is responsible for cleaving the peptide bond between an aspartic acid residue and another amino acid residue in its substrate [56].

NF-κB (nuclear factor kappa-light-chain-enhancer of activated B cells) is a transcription factor that plays a critical role in regulating the immune response, inflammation, and cell survival. The topology of NF-κB can be described as a dimeric protein composed of two subunits, p50 and p65, that are held together by a dimerization domain. The dimerization domain is located at the N-terminus of each subunit and contains a leucine zipper motif, which allows for the formation of a stable dimer. The p50 subunit is composed of an N-terminal Rel homology region (RHR) and a C-terminal DNA-binding domain. The RHR is responsible for dimerization with the p65 subunit, while the DNA-binding domain is responsible for binding to specific DNA sequences in the promoter regions of target genes. The p65 subunit is composed of an N-terminal RHR, a C-terminal transactivation domain, and a central proline-rich region.

The RHR is responsible for dimerization with the p50 subunit, while the transactivation domain is responsible for recruiting coactivators and other transcriptional machinery necessary for gene expression. In the inactive form of NF-κB, the subunits are held together by a family of inhibitors known as IκBs (inhibitor of kappa B). Upon activation, IκBs are phosphorylated and degraded, allowing the NF-κB subunits to translocate to the nucleus and bind to specific DNA sequences, activating the transcription of target genes [56].

The p53 protein is a tetrameric protein with four subunits, including the DNA-binding domain, core domain, and loop-sheet-helix motif. The tetramerization domain is responsible for stabilizing the tetrameric structure. P53 plays a crucial role in DNA binding, transcriptional regulation, cell cycle regulation, and DNA repair. Post-translational modifications, such as phosphorylation, acetylation, and ubiquitination, can alter its stability, DNA-binding affinity, and transcriptional activity. Mutations in the p53 gene are associated with various cancer types, potentially resulting in loss of tumor suppressor function or oncogenic properties [57].

Understanding the topology and ternary structures of these cancer proteins is therefore crucial for development anticancer agent. The results of molecular docking indicated that compound A01 interacted significantly with all three proteins and intercalated DNA strongly. NF kappa was significantly involved in both hydrogen bonding and hydrophobic interactions, with a top docking score of -8.6 kcal/mol, indicating that A01 has the potential to inhibit this protein. These findings are in accordance with in-vitro cytotoxicity assays. In contrast, compound A02 inhibited the P53 protein significantly with a docking score of -5.7 kcal/mol. The binding affinities of these compounds surpassed those of the standard drug cisplatin. Additionally, their binding scores were on par with those of the standard drug doxorubicin. Remarkably, both compounds demonstrated superior affinity against the Kappa protein compared to doxorubicin. For a more comprehensive understanding of the molecular interactions of the standard drugs, please refer to the **S1 File** (**S1 Table**). These results shed light on the anti-cancer potential of compounds A01 and A02 and their underlying mechanism. The docking scores and amino acid residues implicated in the binding interactions of compounds A01 and A02 are shown in Table 4.

**3.3.1 Interpretation of molecular interactions.** The molecular docking analysis of the interaction between Caspase-3 and A01 revealed a significant docking score of -8.0 kcal/mol. The hydrogen bonding residues involved in the interaction were Glu124, Tyr197, and Arg164 with respective hydrogen bond lengths of 3.08 Å, 3.25 Å, and 3.14 Å (Table 4). The hydrophobic interactions residues were Pro201, Glu124, Lys137, Gly125, Tyr197, Tyr195, Val266, Pro201, and Gly125. The bond length of each hydrophobic interaction is provided in Table 4. These results suggest that compound A01 can potentially inhibit the activity of Caspase-3, a key enzyme involved in the initiation of programmed cell death, by forming stable hydrogen bonds with the protein's key residues and forming hydrophobic interactions with other important residues. Therefore, compound A01 can be considered as a potential anti-cancer agent that may aid in the suppression of tumor growth by targeting Caspase-3. Fig 8 is illustrating the putative 2D and 3D binding interactions of A01 with caspase-3.

The analysis of the interaction between Caspase-3 protein and compound A02 revealed comparable bonding and non bonding interactions. The molecular docking score obtained was -5.3 kcal/mol, indicating a moderate binding affinity between the protein and the compound. The hydrogen bonding residues involved in the interaction were Phe252, Ser251, Asn208, and Arg207, with the corresponding hydrogen bond lengths of 3.14, 2.57, 3.19, and 3.05 angstroms, respectively. These residues are important in the active site of Caspase-3, which is the region responsible for its enzymatic activity. The compound also had hydrophobic interactions with Ser249, Phe250, Trp214, and Asp253 residues, which are known to be important for the stability and specificity of protein-ligand interactions. Overall, these findings

**Table 4. The binding interactions observed during molecular docking investigations.**

| Complex | Binding energy (kcal/mol) | Hydrogen bonding | Hydrogen bond length (Å) | Hydrogen bond angle (°) | Hydrophobic interactions residues | Hydrophobic interactions Bond length (Å) |
|---|---|---|---|---|---|---|
| **Caspase-3- A01** | -8.0 | Glu124, Tyr197, Arg164 | 3.08, 3.25, 3.14 | 137.08, 147.09, 133.54 | Pro201, Glu124, Lys137, Gly125, Tyr197, Tyr195, Val266, Pro201, Gly125 | 3.90, 3.95, 3.55, 3.39, 3.90, 3.85, 3.65, 3.23,3. 99 |
| **Caspase-3- A02** | -5.3 | Phe252, Ser251, Asn208, Arg207, | 3.14, 2.57, 3.19, 3.05 | 126.59, 119.30, 100.16, 136.70 | Ser249, Phe250, Trp214, Asp253 | 2.93, 3.76, 3.45, 4.00 |
| **NF-κB- A01** | -8.6 | Glu49, Glu222, Arg260 | 2.90, 2.99, 2.86 | 154.43, 165.75, 117.68 | Thr52, Arg50, Trp258, Gly259, Phe239, Ile224, Ser51 | 4.00, 4.22, 3.90, 3.20, 3.21, 3.89, 4.01 |
| **NF-κB- A02** | -5.0 | Asp141 | 3.14 | 164.90 | Arg140, Gln111, Pro352, Ile354, Tyr181, Phe142, Tyr351 | 3.45, 3.22, 2.98, 2.90, 3.95, 4.00, 4.01 |
| **P53- A01** | -7.8 | Gln23, Gln23, Ile21 | 3.03, 2.82, 2.81 | 151.60, 120.55, 109.49 | Asn17, Ile22, Tyr92, Leu100, Glu89, Asn232, Lys20, Glu13 | 3.82, 3.65, 3.25, 3.81, 3.99, 3.87, 3.21, 3.54 |
| **P53- A02** | -5.7 | Asn17, Glu89, Arg10 | 3.04, 2.83, 3.19 | 110.43, 148.09, 136.03 | Gln23, Ile21, Tyr92 | 3.35, 2.34, 4.08 |
| **DNA-compound A01** | -8.3 | Dt7, Dt8, Dt8 | 3.03, 3.01, 2.71 | 109.21, 134.23, 149.22 | Dt20, Dt19, Da18, Dg10, Da17, Dc9 | 2.54, 2.90, 2.87, 3.00, 3.45, 3.89 |
| **DNA-compound A02** | -5.0 | Dt8, Dt19 | 3.17, 3.07 | 156.22, 124.32 | Dc9, Da18, Da17 | 2.11, 2.09, 2.45 |

suggest that compound A02 has the potential to inhibit the activity of Caspase-3, which is a promising target for anti-cancer drug development. Fig 9 is illustrating the putative binding mode of compound A02 with caspase-3.

The results from molecular docking analysis of compound A01 and NF-κB showed a strong binding affinity with a high docking score of -8.6 kcal/mol. The key residues involved in the formation of hydrogen bonds were Glu49, Glu222, and Arg260, with hydrogen bond lengths ranging from 2.86 to 2.99 angstroms. Furthermore, hydrophobic interactions also played a vital role in the binding of compound A01 (A01) with NF-κB, with the key residues being Thr52, Arg50, Trp258, Gly259, Phe239, Ile224, and Ser51. These investigations provide insight

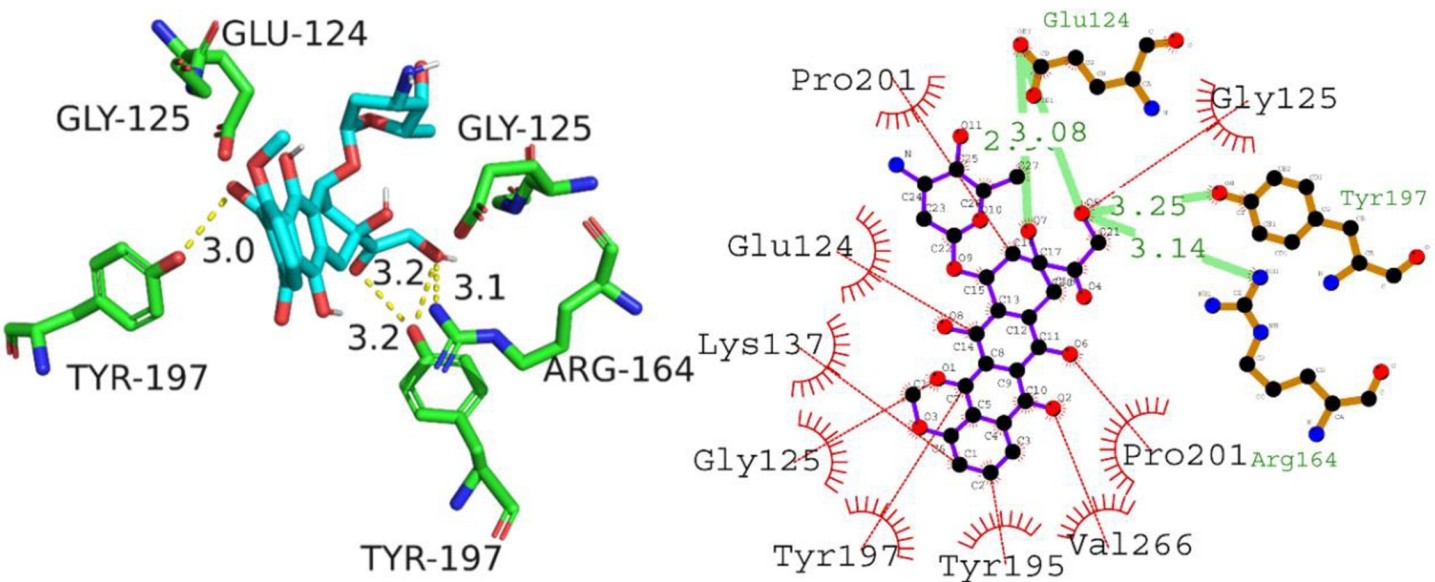

**Fig 8. Predicted 2D and 3D binding interactions of compound A01 with caspase 3.**

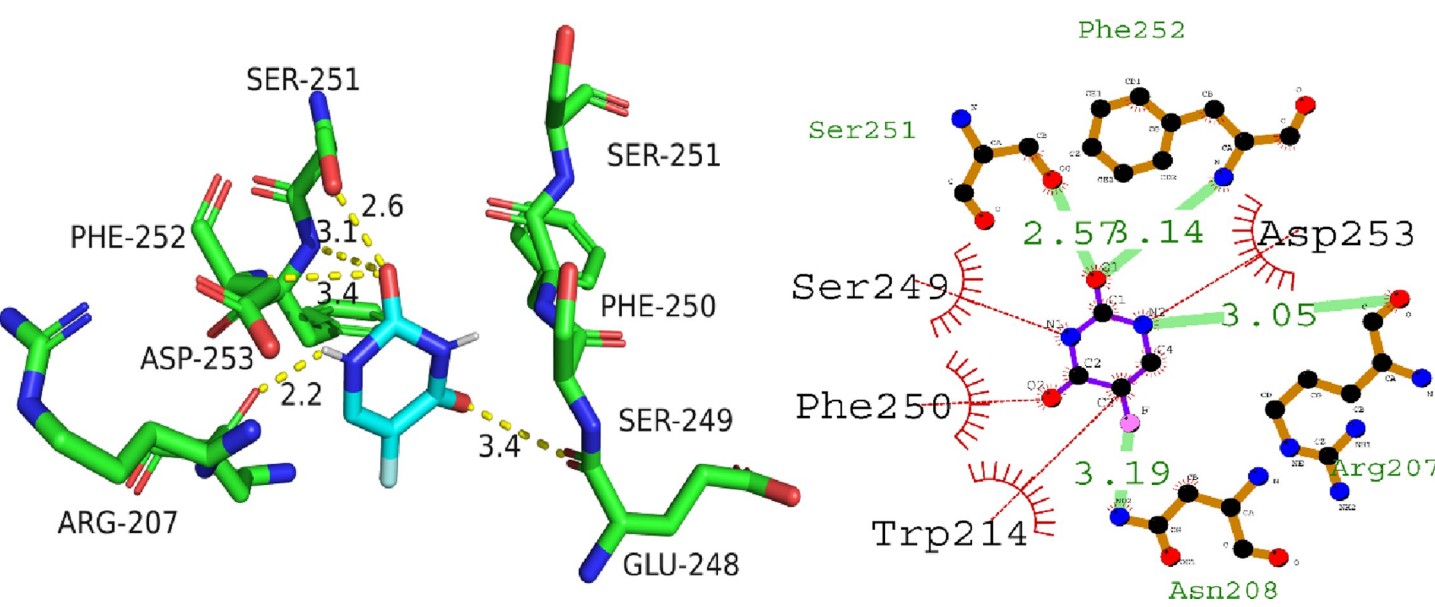

**Fig 9. Putative 2D and 3D binding mode of compound A02 against Caspase-3.**

into the substantial potential of compound A01 as an anti-cancer agent by inhibiting the NF-κB protein. Fig 10 depicts the putative binding mode of compound A01 (A01) with NF-κB.

The molecular docking data for the complex of NF-κB and compound A02 shows an insignificant docking score of -5.0 kcal/mol. The only hydrogen bonding residue involved in the interaction is Asp141, having a hydrogen bond length of 3.14 angstroms. The hydrophobic interactions involved residues Arg140, Gln111, Pro352, Ile354, Tyr181, Phe142, and Tyr351.

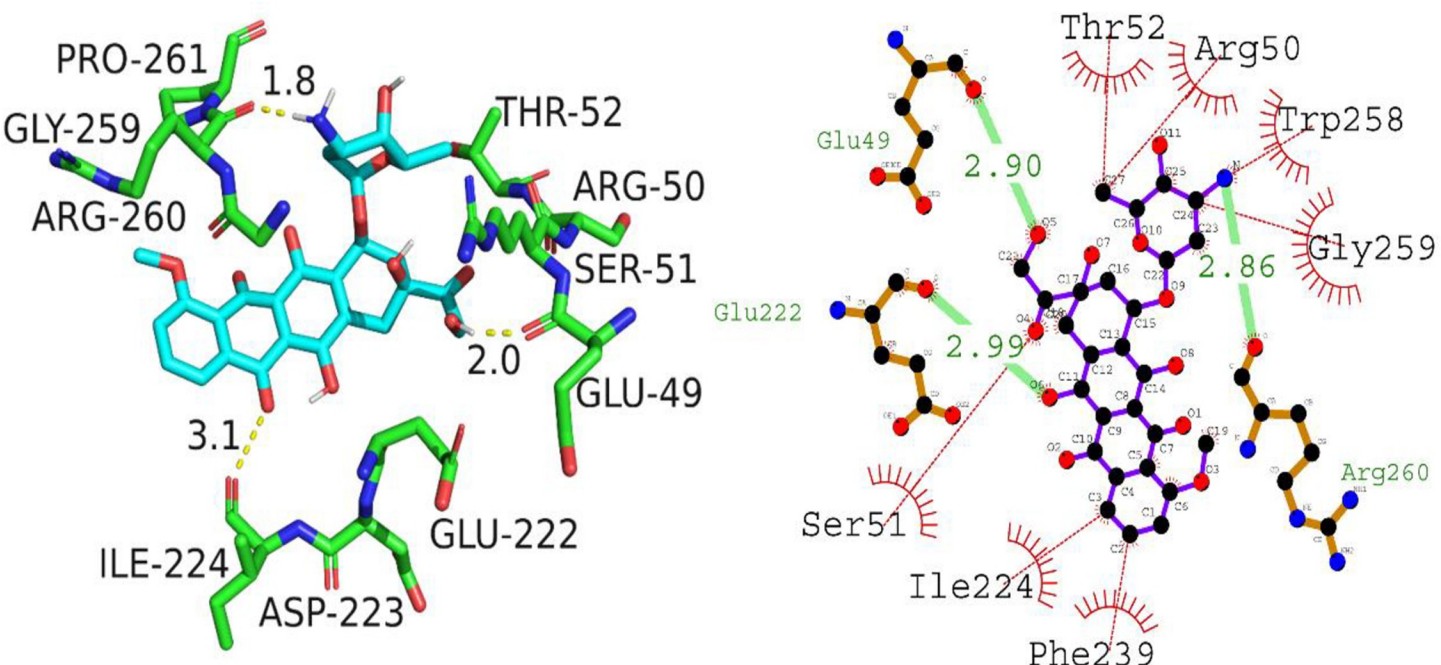

**Fig 10. Putative 2D and 3D interaction of A01 (A01) with NF-κB.**

Overall, the interaction between compound A02 and NF-κB is less favorable than that of compound A01, with a lower docking score and fewer hydrogen bonding and hydrophobic interactions. Fig 11 is presenting 2D and 3D interaction of A02 against NF-κB.

With a docking score of -7.8 kcal/mol, the docking analysis of P53 with A01 revealed a high binding affinity. The interaction between Gln23, Gln23, and Ile21 residues was mediated by hydrogen bonds with lengths ranging from 2.81 to 3.03 angstroms. In addition, hydrophobic interactions were detected between the residues Asn17, Ile22, Tyr92, Leu100, Glu89, Asn232, Lys20, and Glu13. These findings suggest that A01 could function as a tumor suppressant by inhibiting the P53 protein's activity. Fig 12 depicts the orientation of compound A01 binding to P53.

The molecular docking data of P53 with compound A02 revealed comparable docking score of -5.7 kcal/mol. The analysis of the docking results shows that compound A02 forms hydrogen bonds with Asn17, Glu89, and Arg10 residues with bond lengths of 3.04, 2.83, and 3.19 angstroms, respectively. In addition, it interacts with hydrophobic residues such as Gln23, Ile21, and Tyr92. These interactions suggest that compound A02 can effectively bind to the active site of P53 protein and inhibit its activity. Therefore, it can be considered a potential candidate for the development of anti-cancer drugs targeting P53 protein. Fig 13 is illustrating the binding orientation of compound A02 against P53.

**3.3.2 Intercalation of DNA molecule.** In the analysis of the interaction of compound A01 with DNA, the molecular docking data shows that compound A01 strongly intercalated with the DNA molecule, with a docking score of -8.3 kcal/mol. The hydrogen bonding residues involved in this interaction were Dt7, Dt8, and Dt8, with hydrogen bond lengths of 3.03, 3.01, and 2.71 angstroms, respectively. In addition, the compound also showed significant hydrophobic interactions with the DNA molecule through residues Dt20, Dt19, Da18, Dg10, Da17, and Dc9. These findings suggest that compound A01 has a strong potential for anti-cancer activity by interacting with DNA and inhibiting its replication and cell division. Furthermore, the observation that compound A01 is entirely embedded within the DNA helix suggests a robust intercalation of DNA by the compound. Such interactions could hold significant implications for the potential use of compound A01 in treating DNA-related diseases, such as cancer. However, further research is required to gain a comprehensive understanding of the mechanism by which compound A01 interacts with DNA. Fig 14 depicts the intercalation of DNA grooves by A01 (A01).

The intercalation of DNA by compound A02 was evaluated and the results showed a docking score of -5.0 kcal/mol. The compound intercalated between the base pairs of the DNA molecule, specifically between nucleotides Dt8 and Dt19, forming hydrogen bonds with them at a bond length of 3.17 and 3.07 angstroms respectively. Additionally, the compound showed hydrophobic interactions with nucleotides Dc9, Da18, and Da17. Intercalation of compounds between the base pairs of DNA is known to be a crucial mechanism of anti-cancer agents, as it can interfere with DNA replication and lead to cell death. These findings suggest that compound A02 has the potential to act as an anti-cancer agent by inhibiting DNA replication and inducing cell death. Further studies are needed to confirm the efficacy of this compound as a potential anti-cancer agent. Fig 15 is illustrating the intercalation of DNA grove by A02.

## 3.4. Molecular dynamics simulation

This study used molecular dynamics simulations to examine how well compounds A01 and A02 (A02) maintained their optimal conformations when bound to their respective target proteins. Compound A01 (A01) was discovered to have the highest binding affinity for NF-κB, while compound A02 (A02) was discovered to significantly inhibit P53. In addition, the

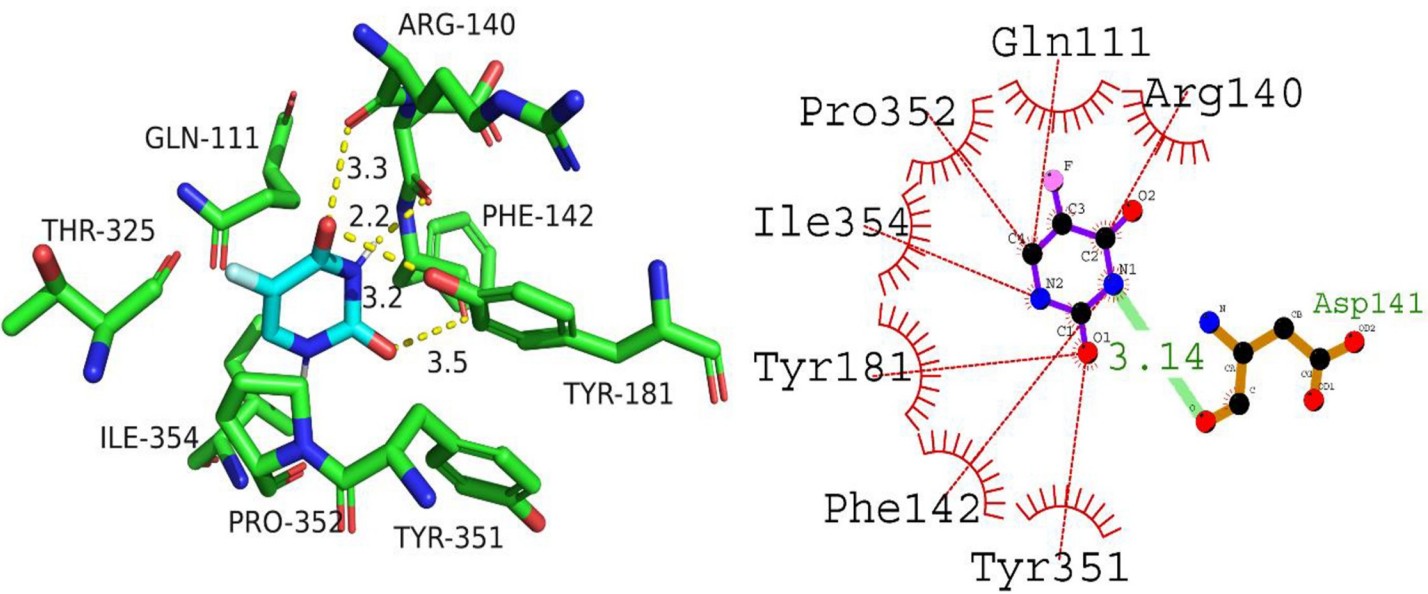

**Fig 11. Illustration of 2D and 3D interaction of A02 against NF-κB.**

compound A01 and A02 in complex with caspase 3 was also subjected to MD simulation and detailed discussion has been incorporated in the **S1 File** (**S3 Fig**). The primary objective of these simulations was to determine how the molecular interactions between the ligand and protein affected the stability of the protein-ligand complex. Protein-ligand complexes were modeled in Desmond and then subjected to mechanical and thermal stresses to ascertain their stability. After accumulating and organizing the data, scientists analyzed it to draw conclusions about the protein-ligand interaction and identify the key residues involved in complex formation.

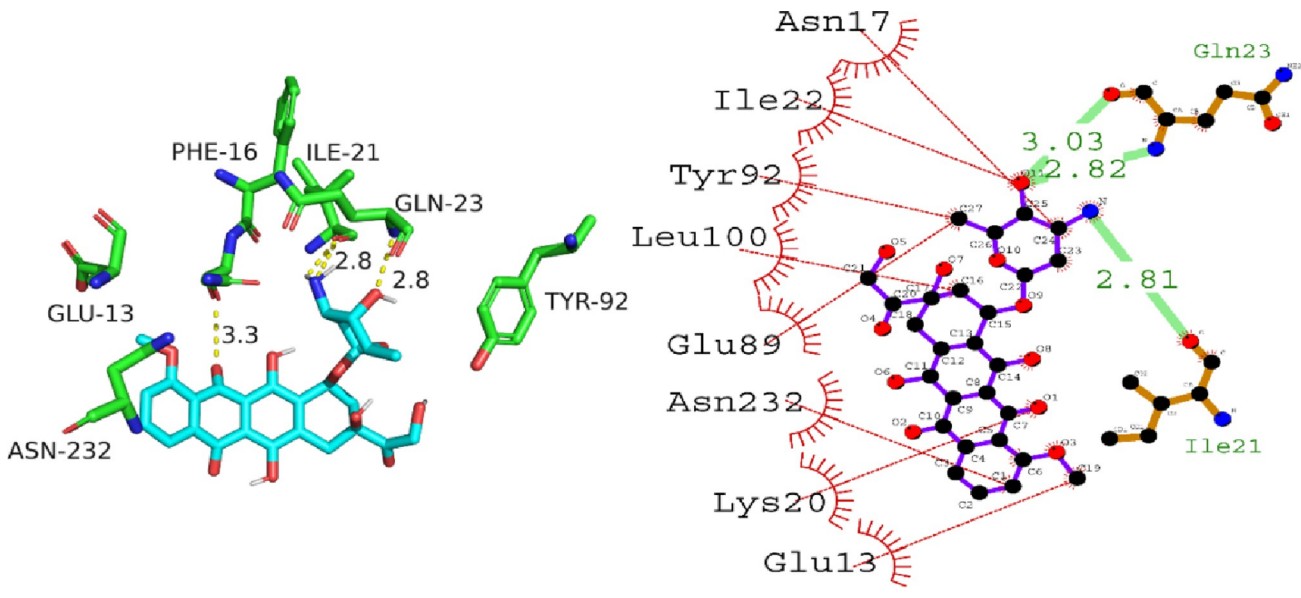

**Fig 12. The putative binding mode of compound A01 against P53.**

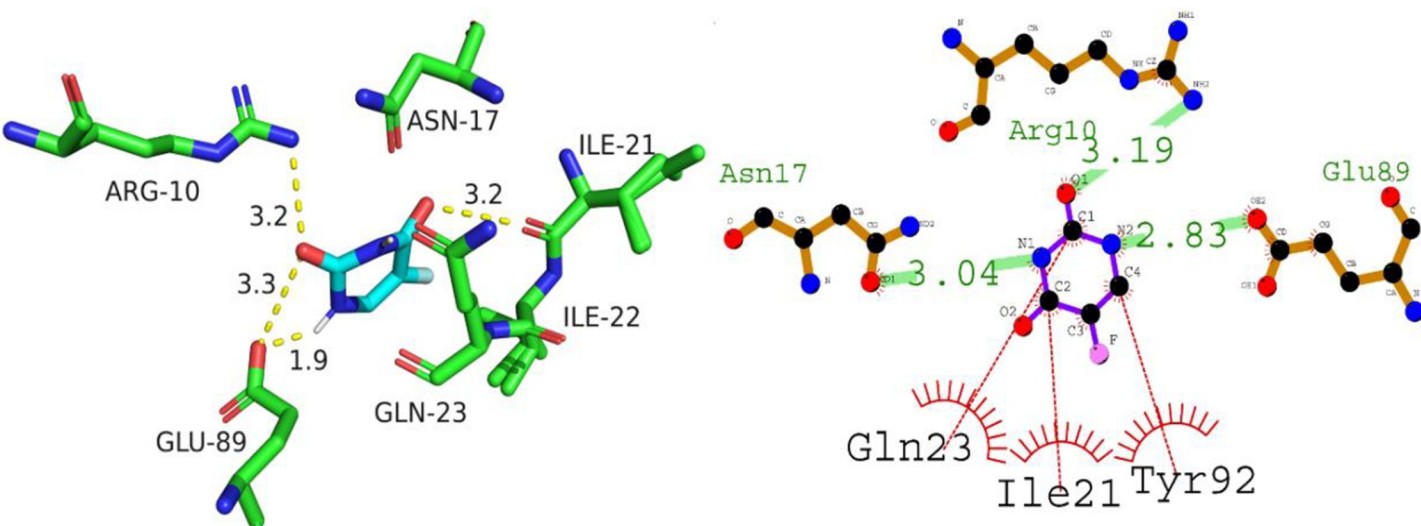

**Fig 13. The putative binding mode of compound A02 against P53.**

This research seeks to analyze RMSD patterns during molecular dynamics simulations in order to gain a better understanding of the stability of both the apo protein and the protein-ligand complex. The apo protein NF-κB exhibited only minor rearrangements after 10 ns of simulation, demonstrating its remarkable stability. Throughout the voyage, the RMSD for the protein-ligand combination (NF-κB-A01 complex) was discovered to be a constant 3.1 angstroms. Hydrogen bonds and hydrophobic interactions were found to be the primary contributors to the stability of the complex. Hydrophobic interactions reduced the exposure of

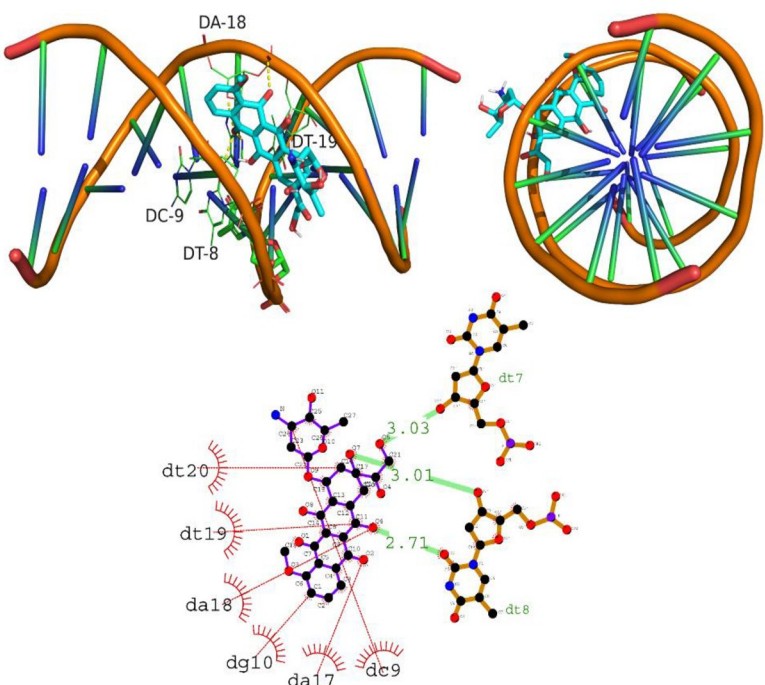

**Fig 14. Intercalation of DNA groove by compound A01.**

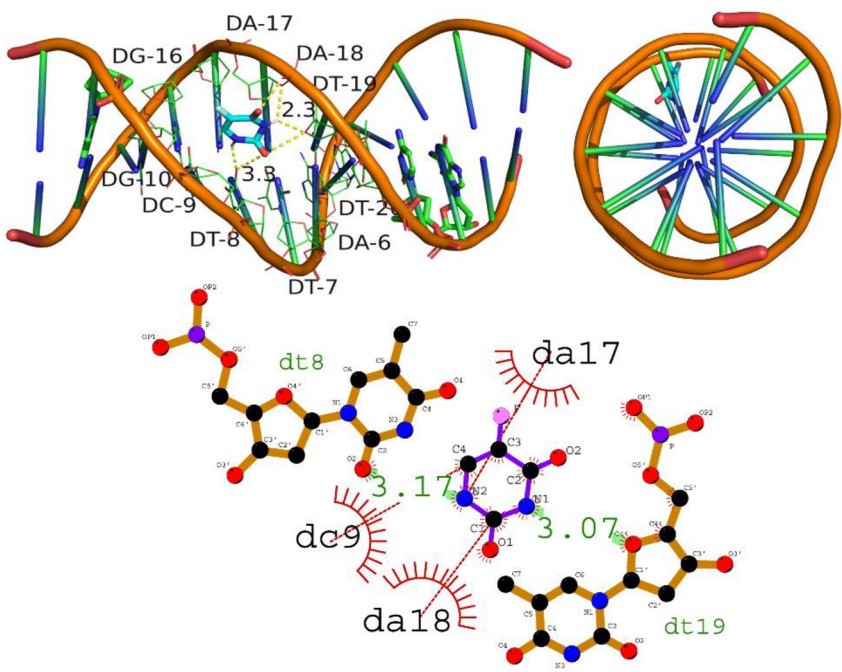

**Fig 15. Intercalation of DNA groove by compound A02.**

hydrophobic residues in both the protein and the ligand, whereas hydrogen bonding residues were essential for complex stability. Fig 16 depicts the evolution of the relative mean squared deviation (RMSD) of both the protein and the protein-ligand complex (NF-κB-A01 complex).

Root Mean Square Fluctuation (RMSF) analysis, which measures the deviation of each amino acid residue from its mean location, was utilized to evaluate the protein-ligand

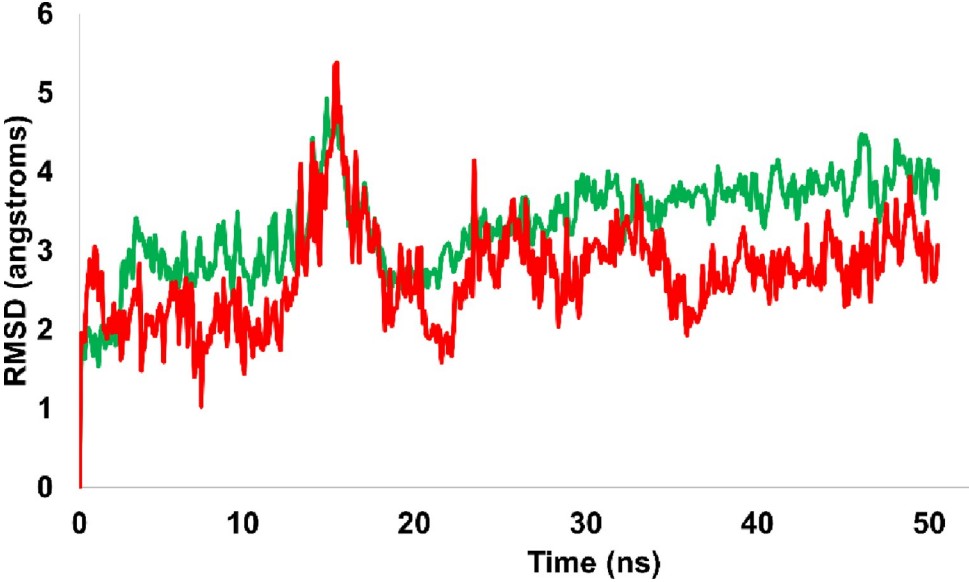

**Fig 16. Representation of RMSD pattern for apo protein NF-κB (red colored trajectory) and liganded protein (NF-κB -A01 complex, represented by green colored trajectory).**

complex's stability. The results demonstrated that the protein-ligand complex did not dissociate during the simulation. The RMSF study confirmed the building's structural integrity and its capacity to maintain its integrity. The investigation shed light on the protein-ligand interaction's stability, an essential property for the complex's application in biological systems. With an average RMSF value of 2.5 angstroms for NF-κB, the study also identified key amino acid residues that remained stable and connected to the ligand. These results provide insight on the significance of a stable protein-ligand complex for proper function, which has implications for the development of novel medications and treatments that target proteins. Fig 17 shows the RMSF values for the intended complex.

The RMSD (Root Mean Square Deviation) analysis was performed on both the apo protein and p53-A02 complex to understand the stability of the simulated trajectories. The apo protein (P53) demonstrated RMSD values ranging from 1.6 to 1.8 angstroms. The apo protein achieved stability at the start of the simulation, and trajectory achieved stability and equilibrated. However, the protein-ligand complex showed slight rearrangements during the simulated trajectory. Initially, up to 10ns, the RMSD of the complex was 3 angstroms, which then increased up to 5 angstroms before eventually dropping and achieving equilibrium. The fluctuations in RMSD values suggest that the protein-ligand complex underwent some conformational changes during the simulation. However, the overall stability of the complex was maintained, and the fluctuations were within an acceptable range. The Fig 18 is illustrating the evolution of RMSD pattern for P53 and P53-A02 complex.

Using RMSF (Root Mean Square Fluctuation), we analyzed the complex's (A02) stability. The majority of the simulated trajectory was stable, with the exception of residues 180–200, which displayed the largest peak in RMSF with a value of up to 2 angstroms. Despite this peak, however, the amino acid residues (10–30) in the binding site that interact with the ligand A02 remained relatively stable throughout the simulation, indicating that the P53-A02 complex is generally stable. The P53-A02 complex is both chemically reactive and stable, as demonstrated by the DFT calculations and MD simulations. Fig 19 shows the RMSF values for the intended complex.

The RMSD pattern for the individual simulated ligands has been comprehensively assessed. The RMSD trajectories of the ligands have unveiled significant conformational changes, offering insights into the accommodation within the ligand pocket. Remarkably, the RMSD values remained consistently below 2 angstroms for both ligands, namely A01 and A02, underscoring their continuous association with the active pocket of the targeted proteins. Furthermore, the ligand-bound complexes displayed convergence and stability, exhibiting minimal conformational alterations. Specifically, the average RMSD for ligand A01 was measured at 1.5 angstroms, while ligand A02 exhibited even greater stability with an average RMSD of 0.8 angstroms. To visually represent the evolutionary trends in RMSD for the individual ligands, refer to Fig 20. This graphical illustration offers a clearer depiction of how the ligands' RMSD values evolved over the simulation period.

Superimposing the initial and final frames would yield valuable insights into the orientation and conformational changes occurring in both the protein and bound ligands. This analysis would also shed light on the conserved and modified ligand-amino acid interactions, along with the close-range contacts that are established or altered during the process. The trajectories of the NF-κB-A01 complex and P53-A02 complex were synchronized at multiple intervals for alignment. Specifically, the initial conformation in the first frame was aligned with the concluding frame of each simulated trajectory. The resulting superimposed configurations visually depicted the progressive RMSD patterns showcased in Figs 13 and 15. It's worth highlighting that throughout this alignment, both ligands consistently maintained their connections with the active sites of the respective target proteins. Minor adjustments within the binding sites

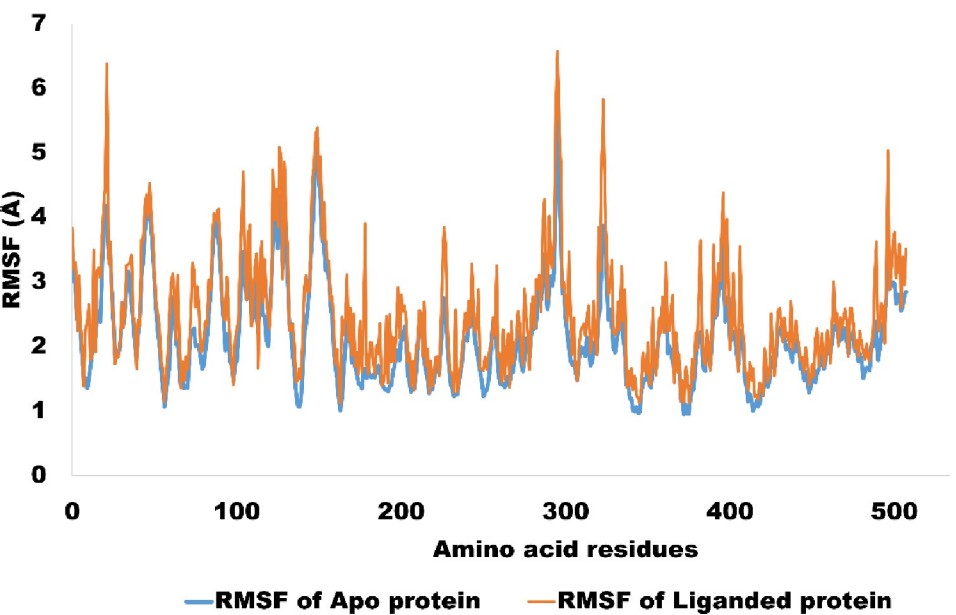

**Fig 17. The amino acid residues wise fluctuation (RMSF) for c alpha atoms of NF-κB (apo protein) and liganded protein (NF-κB -A01 complex).**

were observed, further emphasizing the dynamic nature of the interactions. This alignment process is visually presented in Fig 21, where the superimposed poses of the simulated complexes are illustrated.

**3.4.1 *MMGBSA* analysis.** The MMGBSA free energy calculation is a valuable tool for estimating the binding affinities of protein-ligand complexes. In this study, we employed the Thermal_mmgbsa script of Schrodinger to subject the simulated complexes to MMGBSA

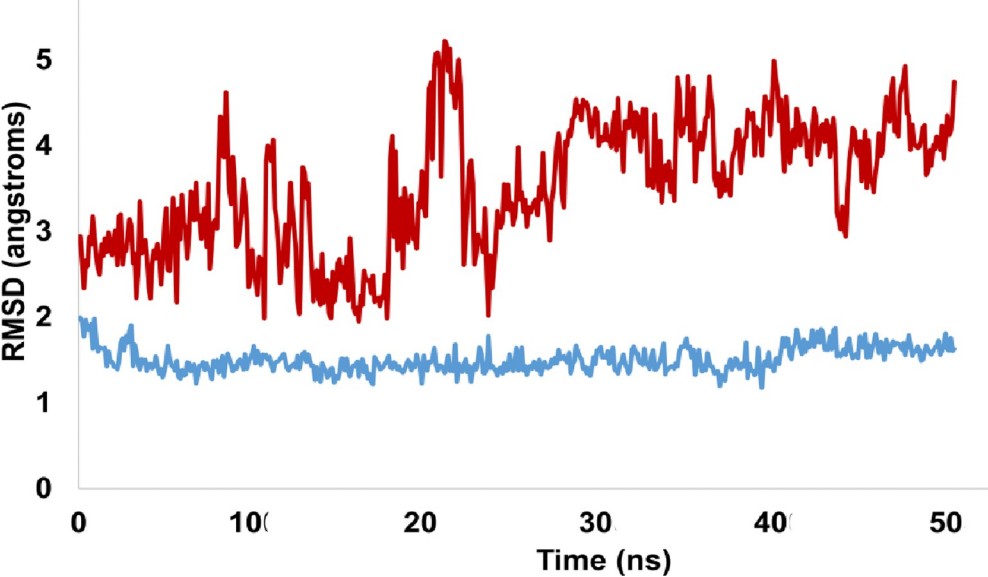

**Fig 18. The RMSD pattern for the apo protein, P53 (represented by the blue trajectory), and the liganded protein, P53-A02 complex (represented by the brown trajectory), is depicted in the illustration.**

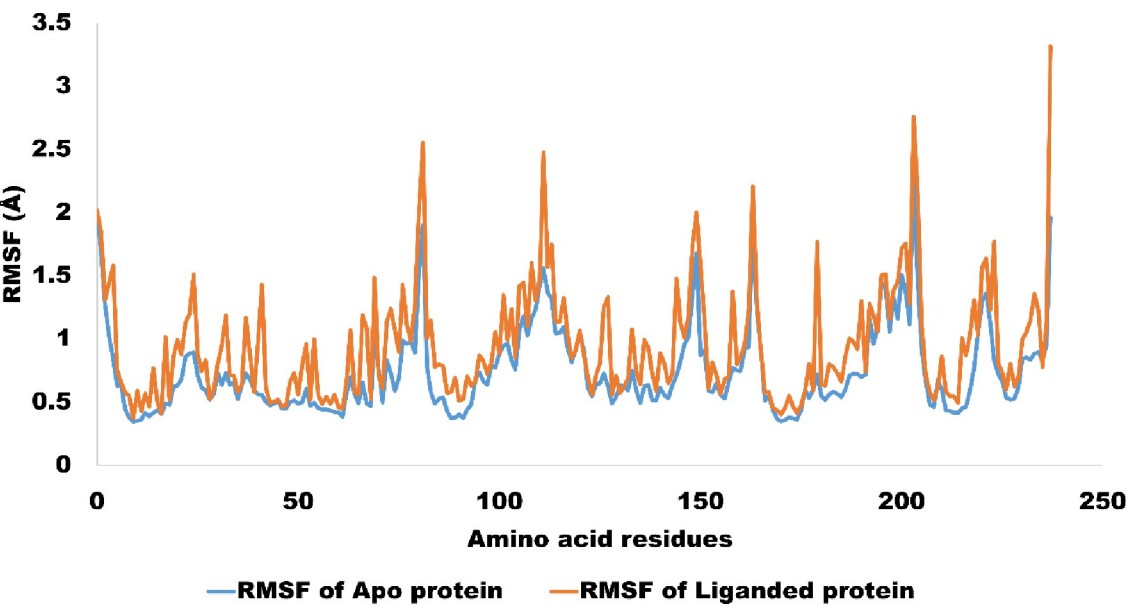

**Fig 19. The amino acid residues wise fluctuation (RMSF) of Apo protein (P53) and liganded protein (P53-A02 complex).**

assay. The resulting MMGBSA free energy calculations for the simulated complexes are presented in Table 5. These findings demonstrate the potential of MMGBSA free energy calculations as a promising approach for predicting binding affinities of protein-ligand complexes.

### 3.5. *ADMET* properties

The physicochemical properties of Compound A01 and Compound A02 were analyzed based on various parameters, including their molecular weight, number of heavy atoms, fraction

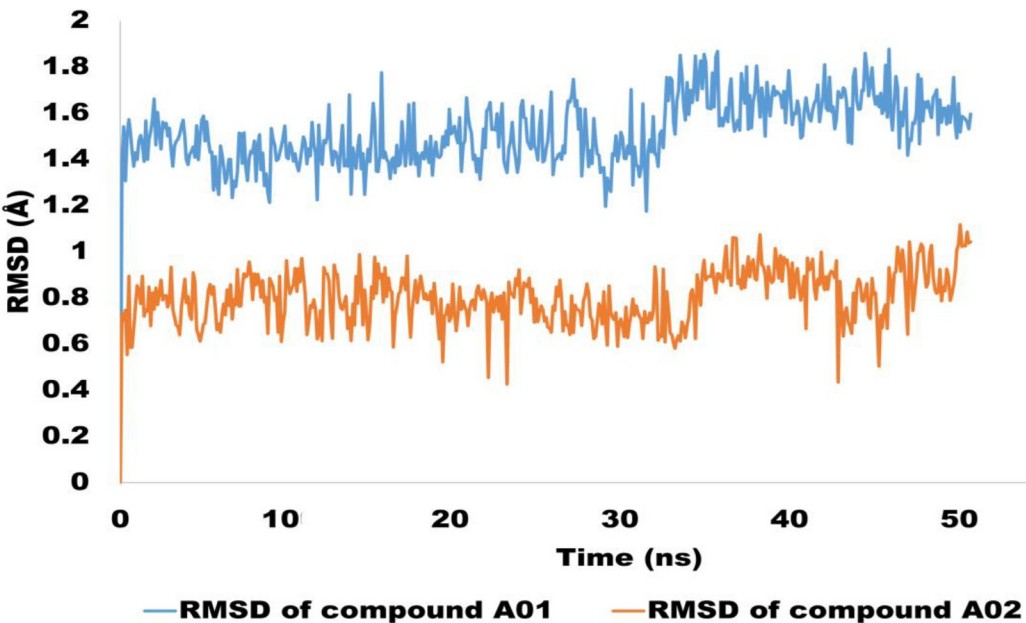

**Fig 20. The evolution of RMSD trajectories for sole ligand molecules.** The Blue colored trajectory is for compound A01 whereas orange colored trajectory is depicting the RMSD for compound A02.

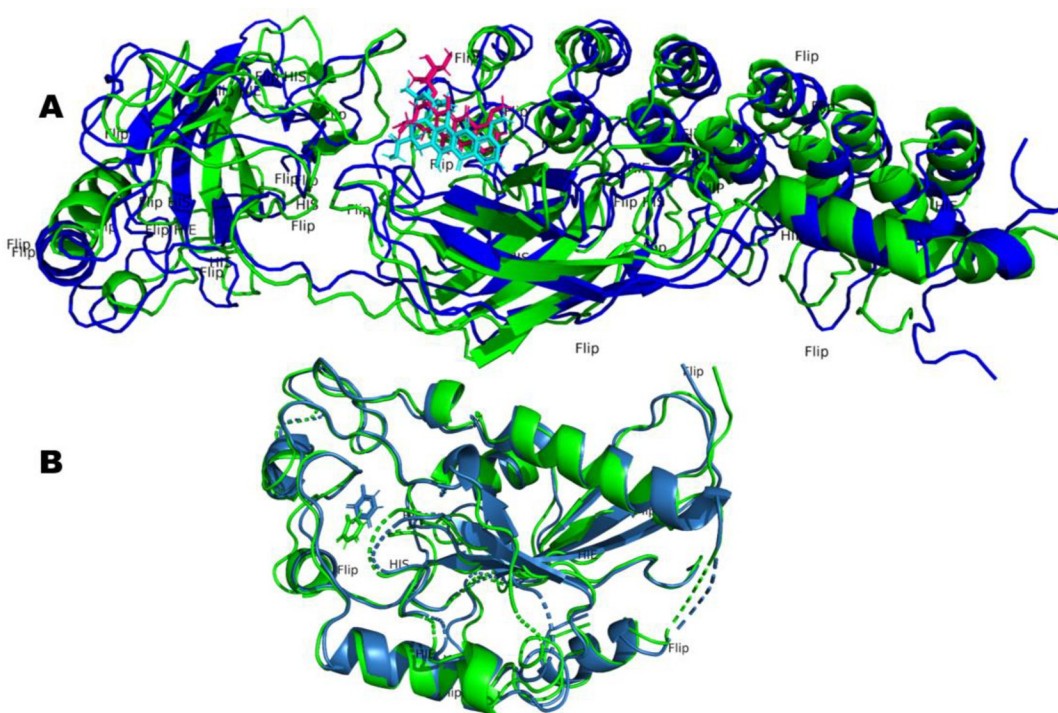

**Fig 21. Superimposed structures of the NF-κB-A01 complex and the P53-A02 complex were observed during MD simulation.** (A) Overlay frames of the NF-κB-A01 complex, where the blue-colored ligand indicates the conformation observed during the first frame, while the pink-colored ligand indicates the conformation at the final frame of the MD simulation. (B) Superimposed conformations of the P53-A02 complex observed during the initial and final frames. The green-colored ligand indicates the conformation of A02 in the first frame, whereas the blue-colored ligand indicates the conformation during the final frame.

Csp3, number of rotatable bonds, and number of H-bond donors and acceptors. Additionally, the analysis included the molar refractivity, TPSA, and several different measures of hydrophobicity, such as iLOGP, XLOGP3, WLOGP, MLOGP, SILICOS-IT, and consensus Log Po/w. The compounds were also evaluated for their GI absorption, BBB permeability, P-gp substrate activity, and CYP1A2 inhibition. Compound A01 has a significantly larger molecular weight (543.52g/mol) compared to Compound A02 (130.08g/mol). This is reflected in the number of heavy atoms, with Compound A01 having 39 heavy atoms as compared to 9 for Compound A02. Compound A01 also has a higher fraction Csp3, indicating a higher proportion of sp3-hybridized carbon atoms. In contrast, Compound A02 has no sp3-hybridized carbon atoms, indicating a fully aromatic structure. Compound A01 has five rotatable bonds, while Compound A02 has none. Additionally, Compound A01 has a higher number of H-bond acceptors and donors, indicating a greater potential for hydrogen bonding interactions. The molar refractivity of Compound A01 is much larger than that of Compound A02, indicating a greater polarizability of the molecule. In terms of hydrophobicity, Compound A01 has a higher iLOGP, XLOGP3,

**Table 5. MM-GBSA binding energies of NF-κB-A01 complex and P53-A02 complex.**

| Compounds | $\Delta G_{bind}$ (kJ/mol) | $\Delta E_{coulomb}$ (kJ/mol) | $\Delta E_{covalent}$ (kJ/mol) | $\Delta E_{H\text{-bond}}$ (kJ/mol) | $\Delta E_{vdW}$ (kJ/mol) | Lipophilic energy (kJ/mol) | Sol_GB (kJ/mol) |
|---|---|---|---|---|---|---|---|
| *NF-κB-A01 complex* | -253.34 | -48.36 | 8.34 | -23.56 | -120.43 | -59.74 | -30.89 |
| *P53-A02 complex* | -154.38 | -40.12 | 9.98 | -11.66 | -109.25 | -45.56 | -26.31 |

and SILICOS-IT values, indicating a greater hydrophobic character. However, the WLOGP and MLOGP values for Compound A01 are negative, indicating that the compound is more hydrophilic. The consensus Log Po/w value for Compound A01 is also positive, indicating a higher overall hydrophobic character. In contrast, Compound A02 has negative values for all measures of hydrophobicity, indicating a hydrophilic character. Compound A01 is predicted to have low GI absorption, while Compound A02 is predicted to have high GI absorption. Neither compound is predicted to be a BBB permeant. Compound A01 is predicted to be a substrate for P-gp, while Compound A02 is not. Neither compound is predicted to be a CYP1A2 inhibitor. Overall, the physicochemical properties of Compound A01 and Compound A02 differ significantly, with Compound A01 having a larger size, higher hydrophobicity, and lower GI absorption and being a P-gp substrate, while Compound A02 is smaller, more hydrophilic, and has higher GI absorption and no P-gp substrate activity. These properties can be used to guide drug design and optimization efforts for these compounds (Table 6).

## 4. Conclusion

The present study is focused on the exploration of anticancer potential of two compounds i.e., (7S,9S)-7-[(2R,4S,5S,6S)-4-amino-5-hydroxy-6-methyloxan-2-yl]oxy-6,9,11-trihydroxy-9-(2-hydroxyacetyl)-4-methoxy-8,10-dihydro-7H-tetracene-5,12-dione (A01) and 5-fluoro-1H-pyrimidine-2,4-dione (A02). The anticancer effect was observed against three cancer cell lines and results were found very promising. Briefly, during DFT study for both compounds, the HOMO and LUMO orbitals were found to be locally confined to specific regions of the molecules, indicating intense chemical reactivity. The binding free energy of compound **A01** is substantially influenced by both electrostatic and non-electrostatic interactions, making it a promising anticancer drug. The effect was observed on different cancer proteins highly expressed in both breast and cervical cancer. According to molecular docking analyses, **A01**

**Table 6. Comprehensive physicochemical properties of compound A01 and A02.**

| Physicochemical properties | Compound A01 | Compound A02 |
|---|---|---|
| Formula | C27H29NO11 | C4H3FN2O2 |
| Molecular weight | 543.52g/mol | 130.08g/mol |
| Number of heavy atoms | 39 | 9 |
| Number aromatic heavy atoms | 12 | 6 |
| Fraction Csp3 | 0.44 | 0.00 |
| Number rotatable bonds | 5 | 0 |
| Number H-bond acceptors | 12 | 3 |
| Number H-bond donors | 6 | 2 |
| Molar Refractivity | 132.66 | 27.64 |
| TPSA | 206.07 Å$^2$ | 65.72 Å$^2$ |
| Log $P_{o/w}$ (iLOGP) | 2.58 | 0.44 |
| Log $P_{o/w}$ (XLOGP3) | 1.27 | -0.89 |
| Log $P_{o/w}$ (WLOGP) | -0.32 | -0.38 |
| Log $P_{o/w}$ (MLOGP) | -2.10 | -0.73 |
| Log $P_{o/w}$ (SILICOS-IT) | 1.17 | 1.78 |
| Consensus Log $P_{o/w}$ | 0.52 | 0.05 |
| GI absorption | Low | High |
| BBB permeant | No | No |
| P-gp substrate | Yes | No |
| CYP1A2 inhibitor | No | No |

has a high affinity for binding to NF-κB, whereas **A02** has a higher affinity for binding to p53. The MD simulations provided additional evidence that both compounds are securely bound to their binding sites. These findings shed light on the therapeutic potential of these compounds as inhibitors of NF-κB and P53. By combining computational and experimental methodologies, novel treatments for a wide spectrum of diseases, such as cancer and its associated malignancies, can be developed.

## Supporting information

**S1 File. Supplementary information includes the experimental, bioactivity protocol, and docking protocol.**
(DOCX)

**S1 Data.**
(ZIP)

**S1 Graphical abstract.**
(DOCX)

## Author Contributions

**Conceptualization:** Mubashir Aziz, Syeda Abida Ejaz, Chen Li.

**Data curation:** Muhammad Sarfraz.

**Formal analysis:** Muhammad Sarfraz, Tasneem Zehra, Chen Li.

**Funding acquisition:** Mosab Arafat.

**Investigation:** Muhammad Khurrum Ibrahim, Tasneem Zehra, Mosab Arafat.

**Methodology:** Mubashir Aziz, Mosab Arafat.

**Project administration:** Hanan A. Ogaly.

**Resources:** Fatimah A. M. Al-Zahrani.

**Supervision:** Syeda Abida Ejaz, Chen Li.

**Validation:** Syeda Abida Ejaz.

**Visualization:** Syeda Abida Ejaz, Hanan A. Ogaly, Fatimah A. M. Al-Zahrani.

**Writing – original draft:** Mubashir Aziz.

**Writing – review & editing:** Muhammad Khurrum Ibrahim, Tasneem Zehra, Fatimah A. M. Al-Zahrani.

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
