## [Decision Letter · Decision Letter 0]

30 Jun 2023

PONE-D-23-17723Evaluation of Anticancer Potential of Tetracene-5,12-Dione (A01) and Pyrimidine-2,4-Dione (A02) via Caspase 3 and Lactate Dehydrogenase cytotoxicity investigationsPLOS ONE

Dear Dr. Ejaz,

Thank you for submitting your manuscript to PLOS ONE. After careful consideration, we feel that it has merit but does not fully meet PLOS ONE’s publication criteria as it currently stands. Therefore, we invite you to submit a revised version of the manuscript that addresses the points raised during the review process.

We look forward to receiving your revised manuscript.

Kind regards,

Ahmed A. Al-Karmalawy, Ph.D.

Academic Editor

PLOS ONE

Journal Requirements:

   "Deanship of Scientific Research (DSR), King Khalid University, Abha, Saudi Arabia, under Grant No. RGP.1/282/43."

   "NO"

   "NO"

7. Please ensure that you refer to Figure 15 in your text as, if accepted, production will need this reference to link the reader to the figure.

Reviewers' comments:

Reviewer's Responses to Questions

**Comments to the Author**

1. Is the manuscript technically sound, and do the data support the conclusions?

Reviewer #1: Yes

Reviewer #2: Partly

Reviewer #3: Partly

2. Has the statistical analysis been performed appropriately and rigorously? 

Reviewer #1: Yes

Reviewer #2: Yes

Reviewer #3: Yes

3. Have the authors made all data underlying the findings in their manuscript fully available?

Reviewer #1: Yes

Reviewer #2: Yes

Reviewer #3: No

4. Is the manuscript presented in an intelligible fashion and written in standard English?

Reviewer #1: Yes

Reviewer #2: Yes

Reviewer #3: Yes

5. Review Comments to the Author

Reviewer #1: I think authors should add the images of the in vitro and the other activities and the graphical abstract for the same. I recommend to add the more references in the manuscript. overall it was good manuscript.

Reviewer #2: The current study has aimed to use pyrimidine and pyrazole-based compounds against Caspase-3 , NF-Kappa-b , and p53 proteins using breast cancer and cervical cancer cell line experiments combined with molecular docking and dynamic simulation analysis. Two compounds were assessed in this study, compound A01, and A02 and they both sound to have good cytotoxic potential. However, there are some serious corrections demanded in the following manuscript:

1. The abstract dose not differentiate clearly between the results of other groups and the results of the current studies.

2. lines :53-55, the sentence in line 53 can have confusing interpretations.” the dysfunctional operation of deoxyribonucleic acid is primarily to blame for its development (DNA)” . It is not ideal to say that dysfunction of DNA is the primarily reason for development of cancer, but differential expression level of certain genes in cancer cells as a result of certain dysfunctional and altered cellular pathways, proteins and transcription factors, have been associated with the progression, differentiation and development of different kinds of cancer. Please provide a related reference for line 53-55.

3. Lines 63-64: “ Caspases 3, p53, and NF-kappa-B are potential therapeutic

targets for malignancy “. The authors should specify that these proteins have been suggested for what kind of cancer? Reference 8 is talking about bladder cancer; can the findings of a single study be implied on all types of cancers in general? Please provide more references that have reported the importance of these proteins in other types of cancer as well.

4. Lines 67-68: the authors have explained about the role of the three targets specified in lines 63-64 in cancer, while each protein has a specific role in apoptotic pathway, cell cycle, DNA repair and response to inflammation, and finally in lines 67-68 it is written that they are possible targets for design of anti-cancer agents. There is two problems with these sentences: first, if the current study aims to target DNA repair pathway, talking about the benefits of other pathways such as inflammation is not beneficial, the authors should try to design inhibitors against a specific set of proteins that are known to function in a certain pathway that is associated with progression of certain types of cancers. Second, the authors should try to define which pathway and target is suitable for design of anti-cancer agents in what type of cancer. Targeting a general pathway might not lead to optimum results in all types of cancer. Therefore, it is important to focus on a specific type or types of cancers and specific pathways for design of new anticancer agents.

5. Lines 86-89, based on reference 19, the three suggested targets could have potential to be investigated for further development of new anticancer inhibitors in cervical cancer and breast cancer. Therefore, in lines 87-89 , it should be specified what type of cancer could be potentially treated if the Caspase3, p53, and NF-B proteins get targeted for inhibition. Based on the cell lines selected for this study, it is suggested to add references that have analyzed the function and role of these proteins in breast cancer and cervical cancer.

6. It is not clear in the introduction either the authors are aiming to target PARP protein or Caspase3, p53, and NF-B factors. References 20 and 23 are claiming that pyrimidine and pyrazole-based compounds have potential to inhibit the EGFR structure. Therefore, the authors should clarify based on references if the pyrimidine and pyrazole-based compounds have the potential to disturb the function of these specified proteins if any previous study is available about it.

7. In method section part 3., the authors should try not to use first-person pronouns “we” frequently.

8. Lines 336-338, the docking results of compounds A01 and A02 with Caspase-3, NF-Kappa-B and p53 does not sound to be significant, docking scores in ranges of -5 kcal/mol to -8 kcal/mol are pretty insignificant and not enough to make a strong conclusion over the binding affinity and potential of these compounds with following proteins and can just be used to make a prediction about their possible binding modes with the structures of specified proteins.

9. Line 442-443, its not clear that the “significant binding affinity” of compound A01 with NF-Kappa-B and A02 with p53 is based on previous docking results or molecular dynamic simulation.

Also, it is better to say that compound A02 was shown to have a relative binding affinity with p53 protein. The sentence “compound A02 (A02) was discovered to significantly inhibit P53” is a bit more exaggeration compared to what the docking results demonstrated.

10. Is figure 14 showing the fluctuations of NF-Kappa-B protein in complex mode with compound A01?

11. Figure 15 should clarify the type of protein used for MD analysis.

12. Why did the authors not perform MD simulation on Caspase-3 with compounds A01 and A02?

13. As the docking results were not significant, the MD simulation is expected to be performed for both compound A01, A02 with all three respective proteins (Caspase-3, NF-KB, p53). There was not such significant gap in the docking results of the following compounds with the selected proteins. Therefore, it not convincing enough to claim compound A01 had higher affinity with NF-KB and A02 with p53 protein and so the authors decided to perform MD simulation only on them.

Reviewer #3: Authors of the manuscript entitled “Evaluation of Anticancer Potential of Tetracene-5,12-Dione (A01) and Pyrimidine-2,4Dione (A02) via Caspase 3 and Lactate Dehydrogenase cytotoxicity investigations” introduced both in vitro and in silico investigation regarding the anti-cancer activity of two compounds against different cancerous biotargets. Despite promising findings, the manuscript requires further improvements within several aspects as being highlighted within the provided suggestions and comments:

1. Authors focused on two compounds for their whole study without providing proper introduction (even no actual figure of both compounds except 3D representation within the docking studies). Moreover, the rational for adopting these two particular compounds for investigation is not clear. why among other reported compounds the authors chose these two particular compounds. a better explanation should be highlighted.

2. Within the in vitro cytotoxicity activity assay, the rational for adopting these three cancerous cell line should be highlighted. Whether the authors used these cells for extensive target expression or based on reported citation, either of which should be stated.

3. In line 204, authors should be consistent regarding the annotation of the compounds' cytotoxic activity, either use IC50 or GI50.

4. In sections 3.1.2 and 3.1.3, Levels of statistical significance should be presented within the context and at Figures 2 and 3. Similar to figure 3, the LDH analysis for the control samples should also be represented in Figure 2.

5. IC50 graphs for the investigated compounds should be presented within the supplementary data.

6. Regarding the DFT analysis, authors should provide potential energy surface (PES) scan to allocate the most energetically stabilized geometries. this can be achieved by varying a dihedral angle rotation from 0° to 360° in 10° stepwise rotation while allowing the rest of the molecule to undergo free iterations.

7. The PDB file of both NF-Kappa B and TP53 are without co-crystallized bound ligand. Authors should provide a rational for adopting a particular binding site where molecular modelling simulation were performed.

8. Superimposed binding modes of co-crystallized ligand and their respective redocked poses should be presented at least within the supplementary data.

9. Within the docking results section, author should provide brief description regarding the topology and protein ternary structure of the adopted biotargets prior introducing the docking results. Providing valuable information regarding binding site surface and reported key binding/catalytic residues would help readers to track docking findings.

10. Authors provided comparative data regarding ligand binding modes through both highlighted polar hydrogen bonds and hydrophobic contacts. However, hydrogen binding should be presented within hydrogen bond distances as well as bond angles since hydrogen bond depend on both. Authors should mention the Hydrogen bond angles, since the strength of hydrogen bonding is based on both parameters in a way to ensure the adequacy of optimum hydrogen bonding. Additionally, the authors should mention the distances for the Hydrophobic interactions.

11. In addition to protein’s RMSD, the authors should provide RMSD trajectories for the sole simulated ligands furnishing information regarding ligand-pocket accommodation (ligand was maintained within the pocket or not) as well as relevant ligand's conformational changes as time evolves across the simulation time. Moreover, the sole ligand RMSDs would confirm protein convergence and system stability at the end of the simulation runs in case of ligands' RMSD never exceeds 2-fold the RMSDs of their bounded protein.

12. Overlay of the initial and final frames would provide insights regarding the orientation/conformation changes for both the protein and bounded ligands as well as the conserved and reformed ligand-amino acid bindings and close-range contacts.

13. Notably, Figure 15 showed the apo RMSDs being steadier and with minimal fluctuations as compared to those of the liganded/holo TP53 protein. The latter would question the stability of A02 at the target binding site. Analysis of the above described sole ligand RMSDs as well as overlaid trajectories would speculate such assumptions.

14. Through the RMSF analysis, authors should illustrate trajectories for apo protein as well. This approach would better highlight the impact of compound’s binding on target through pinpointing flexible and immobile patterns for the protein ternary structures and amino acids in reference to the unliganded form. Difference RMSF (ΔRMSF = RMSFApo-Holo) could also be adopted (please refer to doi: 10.1016/j.bmcl.2019.02.031 and doi: 10.1016/j.bmcl.2019.02.031).

15. Findings from the MM-GBSA free binding energy calculations as well as the constituting energy terms (VDW, electrostatic, solvation, SASA) should be represented in Figure or Table for better tracking of binding affinity results.

16. At least in silico ADME_TOX analysis for investigated compounds should be provided through exploring the drug-likeness profile and several pharmacokinetic parameters of these investigated compounds. Findings would be valuable for guiding future drug optimization & development strategies

17. Authors should elaborate more on the discussion section through presenting comparative findings from reported literature studies that investigated other close related compounds against target proteins.

18. Finally, within the discussion sections, authors should highlight the takeaway messages that would be adopted in future lead optimization and development base on the docking and molecular dynamics studies. Prospective/recommended structure modifications to improve the compound’s binding and interactions, as well as pharmacokinetics should be provided within the discussion and conclusion sections.

6. PLOS authors have the option to publish the peer review history of their article (what does this mean?). If published, this will include your full peer review and any attached files.

Reviewer #1: No

Reviewer #2: No

Reviewer #3: No

---

## [Author Response · Author response to Decision Letter 0]

15 Sep 2023

Review Comments to the Author

Reviewer #1: I think authors should add the images of the in vitro and the other activities and the graphical abstract for the same. I recommend to add the more references in the manuscript. overall it was good manuscript.

Response: Graphical abstract and more references have been added in the manuscript. Suggested correction has been made.

Reviewer #2: The current study has aimed to use pyrimidine and pyrazole-based compounds against Caspase-3, NF-Kappa-b, and p53 proteins using breast cancer and cervical cancer cell line experiments combined with molecular docking and dynamic simulation analysis. Two compounds were assessed in this study, compound A01, and A02 and they both sound to have good cytotoxic potential. However, there are some serious corrections demanded in the following manuscript:

1. The abstract dose not differentiate clearly between the results of other groups and the results of the current studies.

Response: We acknowledge the reviewer's concern and have revised the abstract to provide a more distinct separation between the results of other groups and the original contributions of our study.

2. lines :53-55, the sentence in line 53 can have confusing interpretations.” the dysfunctional operation of deoxyribonucleic acid is primarily to blame for its development (DNA)” . It is not ideal to say that dysfunction of DNA is the primarily reason for development of cancer, but differential expression level of certain genes in cancer cells as a result of certain dysfunctional and altered cellular pathways, proteins and transcription factors, have been associated with the progression, differentiation and development of different kinds of cancer. Please provide a related reference for line 53-55.

Response: The suggested changes have been made and reference has been also incorporated.

3. Lines 63-64: “ Caspases 3, p53, and NF-kappa-B are potential therapeutic

targets for malignancy “. The authors should specify that these proteins have been suggested for what kind of cancer? Reference 8 is talking about bladder cancer; can the findings of a single study be implied on all types of cancers in general? Please provide more references that have reported the importance of these proteins in other types of cancer as well.

Response: The role of each protein is elaborated in the manuscript and more references have been incorporated as suggested by the reviewer.

4. Lines 67-68: the authors have explained about the role of the three targets specified in lines 63-64 in cancer, while each protein has a specific role in apoptotic pathway, cell cycle, DNA repair and response to inflammation, and finally in lines 67-68 it is written that they are possible targets for design of anti-cancer agents. There is two problems with these sentences: first, if the current study aims to target DNA repair pathway, talking about the benefits of other pathways such as inflammation is not beneficial, the authors should try to design inhibitors against a specific set of proteins that are known to function in a certain pathway that is associated with progression of certain types of cancers. Second, the authors should try to define which pathway and target is suitable for design of anti-cancer agents in what type of cancer. Targeting a general pathway might not lead to optimum results in all types of cancer. Therefore, it is important to focus on a specific type or types of cancers and specific pathways for design of new anticancer agents.

Response: In current study, we have targeted multiple cell lines including Hela, MDA-MB-231 and MCF7. These cell lines have different level of expression of each proteins including p53, kappa and caspase 3. The selected compounds A01 and A02 were not previously investigated against these cell lines and there anticancer potential is unknown so we have targeted multiple proteins and cell lines via in-silico and in vitro approaches. It was revealed that both compound revealed potential inhibitory activities against all three cell lines. The further mechanistic insight was gained through in-silico studies which had revealed excellent binding affinity with kappa and caspase pathway. In addition, both compounds possessed DNA intercalating potential. Based on these findings, it can be deduced that selected compounds had stronger affinity towards kappa and caspase 3 pathways and possessed strong DNA intercalating activities suggesting their anticancer potential. 

5. Lines 86-89, based on reference 19, the three suggested targets could have potential to be investigated for further development of new anticancer inhibitors in cervical cancer and breast cancer. Therefore, in lines 87-89 , it should be specified what type of cancer could be potentially treated if the Caspase3, p53, and NF-B proteins get targeted for inhibition. Based on the cell lines selected for this study, it is suggested to add references that have analyzed the function and role of these proteins in breast cancer and cervical cancer.

Response: As suggested, the cancer type has been specified and references have been incorporated in the manuscript.

6. It is not clear in the introduction either the authors are aiming to target PARP protein or Caspase3, p53, and NF-B factors. References 20 and 23 are claiming that pyrimidine and pyrazole-based compounds have potential to inhibit the EGFR structure. Therefore, the authors should clarify based on references if the pyrimidine and pyrazole-based compounds have the potential to disturb the function of these specified proteins if any previous study is available about it.

Response: The current study was aimed to target caspase3, p53 and NF-B. The ambiguity related to PARP and EGFR have been omitted from the manuscript and references have been revised.

7. In method section part 3, the authors should try not to use first-person pronouns “we” frequently.

Response: The corrections have been made in entire methodology section.

8. Lines 336-338, the docking results of compounds A01 and A02 with Caspase-3, NF-Kappa-B and p53 does not sound to be significant, docking scores in ranges of -5 kcal/mol to -8 kcal/mol are pretty insignificant and not enough to make a strong conclusion over the binding affinity and potential of these compounds with following proteins and can just be used to make a prediction about their possible binding modes with the structures of specified proteins.

Response: The molecular docking score was considered as significant in comparison to standard drug cisplatin and doxorubicin. The docking score of standard drugs have been incorporated in the supplementary file (Table S1) which provide direct comparison with compound under investigation. In addition, the results were further validated by in vitro analysis.

9. Line 442-443, its not clear that the “significant binding affinity” of compound A01 with NF-Kappa-B and A02 with p53 is based on previous docking results or molecular dynamic simulation.

Response: It was typo error which has been omitted. However compound A01 revealed docking score of -8.6 kcal/mol against kappa-B which is better than compound A02.

Also, it is better to say that compound A02 was shown to have a relative binding affinity with p53 protein. The sentence “compound A02 (A02) was discovered to significantly inhibit P53” is a bit more exaggeration compared to what the docking results demonstrated.

Response: Thankyou anonymous reviewer for deep insight, we have corrected the phrases as per suggestions.

10. Is figure 14 showing the fluctuations of NF-Kappa-B protein in complex mode with compound A01?

Response: Yes, figure 14 is illustrating the RMSF of kappa protein in complex with A01. The caption of figure 14 has been revised.

11. Figure 15 should clarify the type of protein used for MD analysis.

Response : The correction has been made.

12. Why did the authors not perform MD simulation on Caspase-3 with compounds A01 and A02?

Response: Thankyou reviewer for deep insight, the idea behind the simulation of A01 and A01 against NF-KB and p53 was top ranked conformations and high binding affinity. However, we have incorporated the MD simulation of A01 and A02 against caspase 3 in supplementary file.

13. As the docking results were not significant, the MD simulation is expected to be performed for both compound A01, A02 with all three respective proteins (Caspase-3, NF-KB, p53). There was not such significant gap in the docking results of the following compounds with the selected proteins. Therefore, it not convincing enough to claim compound A01 had higher affinity with NF-KB and A02 with p53 protein and so the authors decided to perform MD simulation only on them.

Response: The MD simulation of A01 and A02 against caspase 3 is incorporated in the supplementary file.

Reviewer #3: Authors of the manuscript entitled “Evaluation of Anticancer Potential of Tetracene-5,12-Dione (A01) and Pyrimidine-2,4Dione (A02) via Caspase 3 and Lactate Dehydrogenase cytotoxicity investigations” introduced both in vitro and in silico investigation regarding the anti-cancer activity of two compounds against different cancerous biotargets. Despite promising findings, the manuscript requires further improvements within several aspects as being highlighted within the provided suggestions and comments:

1. Authors focused on two compounds for their whole study without providing proper introduction (even no actual figure of both compounds except 3D representation within the docking studies). Moreover, the rational for adopting these two particular compounds for investigation is not clear. Why among other reported compounds the authors chose these two particular compounds. a better explanation should be highlighted.

Response: The actual structure of both compounds have been incorporated in the figure 1 and rationale for selecting both derivatives has been incorporated in the manuscript.

2. Within the in vitro cytotoxicity activity assay, the rational for adopting these three cancerous cell line should be highlighted. Whether the authors used these cells for extensive target expression or based on reported citation, either of which should be stated.

Response: The primary objective of this study was to explore promising leads for combatting specific types of cancer, particularly breast and cervical cancers. Extensive research has established MCF-7 and MDA-MB-231 as breast cancer cell lines, while the HeLa cell line is associated with cervical metastasis. Strikingly, compound A01 exhibited robust inhibition across all three cell lines, demonstrating its potential as a dual inhibitor for both forms of cancer. The same discussion has been added in the manuscript. 

3. In line 204, authors should be consistent regarding the annotation of the compounds' cytotoxic activity, either use IC50 or GI50.

Response: The correction has been made.

4. In sections 3.1.2 and 3.1.3, Levels of statistical significance should be presented within the context and at Figures 2 and 3. Similar to figure 3, the LDH analysis for the control samples should also be represented in Figure 2.

Response: Figure have been updated as suggested.

5. IC50 graphs for the investigated compounds should be presented within the supplementary data.

Response: The GI50 graphs have been provided in the supplementary file as suggested (Figure S1). 

6. Regarding the DFT analysis, authors should provide potential energy surface (PES) scan to allocate the most energetically stabilized geometries. This can be achieved by varying a dihedral angle rotation from 0° to 360° in 10° stepwise rotation while allowing the rest of the molecule to undergo free iterations.

Response: As per suggestion of the reviewer, we have incorporated the ESP map of both ligands in the revised manuscript.

7. The PDB file of both NF-Kappa B and TP53 are without co-crystallized bound ligand. Authors should provide a rational for adopting a particular binding site where molecular modelling simulation were performed.

Response: We conducted an in-depth literature review to identify proposed binding sites for the target proteins. Moreover, we meticulously evaluated the functional significance of the chosen binding sites. Our analysis drew from a comprehensive range of literature sources to pinpoint these specific binding sites for both proteins. Literature reference is given below; 

Wani TA, Zargar S. Molecular Spectroscopy Evidence of 1, 3, 5-Tris (4-carboxyphenyl) benzene Binding to DNA: Anticancer Potential along with the Comparative Binding Profile of Intercalation via Modeling Studies. Cells. 2023 Apr 10; 12(8):1120.

8. Superimposed binding modes of co-crystallized ligand and their respective redocked poses should be presented at least within the supplementary data.

Response: The suggested data has been incorporated in the supplementary file. RMSD of less than 2 angstroms between native and redocked pose reflects the validation of docking protocol.

9. Within the docking results section, author should provide brief description regarding the topology and protein ternary structure of the adopted bio targets prior introducing the docking results. Providing valuable information regarding binding site surface and reported key binding/catalytic residues would help readers to track docking findings.

Response: The description related to topology and ternary structure of all bio targets have been incorporated in the manuscript. 

10. Authors provided comparative data regarding ligand binding modes through both highlighted polar hydrogen bonds and hydrophobic contacts. However, hydrogen binding should be presented within hydrogen bond distances as well as bond angles since hydrogen bond depend on both. Authors should mention the Hydrogen bond angles, since the strength of hydrogen bonding is based on both parameters in a way to ensure the adequacy of optimum hydrogen bonding. Additionally, the authors should mention the distances for the hydrophobic interactions.

Response: The suggested data including hydrogen bond angle and hydrophobic interactions bond length have been incorporated in the table 4.

11. In addition to protein’s RMSD, the authors should provide RMSD trajectories for the sole simulated ligands furnishing information regarding ligand-pocket accommodation (ligand was maintained within the pocket or not) as well as relevant ligand's conformational changes as time evolves across the simulation time. Moreover, the sole ligand RMSDs would confirm protein convergence and system stability at the end of the simulation runs in case of ligands' RMSD never exceeds 2-fold the RMSDs of their bounded protein.

Response: As suggested by the reviewer, the RMSD pattern for each sole ligand has been incorporated in the manuscript. Both ligand remained significantly stable with RMSD less than 1.5 angstorms. The detailed discussion has been incorporated in the manuscript.

12. Overlay of the initial and final frames would provide insights regarding the orientation/conformation changes for both the protein and bounded ligands as well as the conserved and reformed ligand-amino acid bindings and close-range contacts.

Response: Super imposed snapshots of MD simulated trajectory at various intervals have been incorporated in the manuscript. 

13. Notably, Figure 15 showed the apo RMSDs being steadier and with minimal fluctuations as compared to those of the liganded/holo TP53 protein. The latter would question the stability of A02 at the target binding site. Analysis of the above described sole ligand RMSDs as well as overlaid trajectories would speculate such assumptions.

Response: The RMSD of solo ligand and snapshots of overlaid trajectories at initial, and final frame has been incorporated in the manuscript which is vanishing the speculations.

14. Through the RMSF analysis, authors should illustrate trajectories for apo protein as well. This approach would better highlight the impact of compound’s binding on target through pinpointing flexible and immobile patterns for the protein ternary structures and amino acids in reference to the unliganded form. Difference RMSF (ΔRMSF = RMSFApo-Holo) could also be adopted (please refer to doi: 10.1016/j.bmcl.2019.02.031 and doi: 10.1016/j.bmcl.2019.02.031).

Response: Thankyou for suggesting impactful revisions, we have made suggested changes and due citation has been made. RMSF of apo as well as liganded protein has been incorporated in the manuscript.

15. Findings from the MM-GBSA free binding energy calculations as well as the constituting energy terms (VDW, electrostatic, solvation, SASA) should be represented in Figure or Table for better tracking of binding affinity results.

Response: The suggested data has been added in the revised version (table 5).

16. At least in silico ADME_TOX analysis for investigated compounds should be provided through exploring the drug-likeness profile and several pharmacokinetic parameters of these investigated compounds. Findings would be valuable for guiding future drug optimization & development strategies

Response: The ADME_TOX of both compounds has been calculated through Swiss ADME tool and detailed discussion has been incorporated in the manuscript.

17. Authors should elaborate more on the discussion section through presenting comparative findings from reported literature studies that investigated other close related compounds against target proteins.

Response: The discussion part has been revised as suggested

18. Finally, within the discussion sections, authors should highlight the takeaway messages that would be adopted in future lead optimization and development base on the docking and molecular dynamics studies. Prospective/recommended structure modifications to improve the compound’s binding and interactions, as well as pharmacokinetics should be provided within the discussion and conclusion sections.

Response: The suggested data has been provided as suggested. We are thankful for the reviewer suggestions and we have done the extensive revision of this manuscript by keeping in view the aim of the study.

---

## [Decision Letter · Decision Letter 1]

21 Sep 2023

Evaluation of Anticancer Potential of Tetracene-5,12-Dione (A01) and Pyrimidine-2,4-Dione (A02) via Caspase 3 and Lactate Dehydrogenase cytotoxicity investigations

PONE-D-23-17723R1

Dear Dr. Ejaz,

We’re pleased to inform you that your manuscript has been judged scientifically suitable for publication and will be formally accepted for publication once it meets all outstanding technical requirements.

Kind regards,

Ahmed A. Al-Karmalawy, Ph.D.

Academic Editor

PLOS ONE

Reviewers' comments:

Reviewer's Responses to Questions

**Comments to the Author**

1. If the authors have adequately addressed your comments raised in a previous round of review and you feel that this manuscript is now acceptable for publication, you may indicate that here to bypass the “Comments to the Author” section, enter your conflict of interest statement in the “Confidential to Editor” section, and submit your "Accept" recommendation.

Reviewer #2: All comments have been addressed

Reviewer #3: (No Response)

2. Is the manuscript technically sound, and do the data support the conclusions?

Reviewer #2: Yes

Reviewer #3: Yes

3. Has the statistical analysis been performed appropriately and rigorously? 

Reviewer #2: Yes

Reviewer #3: Yes

4. Have the authors made all data underlying the findings in their manuscript fully available?

Reviewer #2: Yes

Reviewer #3: Yes

5. Is the manuscript presented in an intelligible fashion and written in standard English?

Reviewer #2: Yes

Reviewer #3: Yes

6. Review Comments to the Author

Reviewer #2: All comments and concerns mentioned for the authors have been addressed accordingly very well as suggested.

Reviewer #3: Authors adequately provided responses and committed manuscript modifications as per recommendations and suggestions.

7. PLOS authors have the option to publish the peer review history of their article (what does this mean?). If published, this will include your full peer review and any attached files.

Reviewer #2: No

Reviewer #3: **Yes**

---

## [Editor Report · Acceptance letter]

28 Sep 2023

PONE-D-23-17723R1 

Evaluation of Anticancer Potential of Tetracene-5,12-Dione (A01) and Pyrimidine-2,4-Dione (A02) via Caspase 3 and Lactate Dehydrogenase cytotoxicity investigations 

Dear Dr. Ejaz:

I'm pleased to inform you that your manuscript has been deemed suitable for publication in PLOS ONE. Congratulations! Your manuscript is now with our production department. 

Kind regards, 

on behalf of

Dr. Ahmed A. Al-Karmalawy 

Academic Editor

PLOS ONE